# PERIA: Perceive, Reason, Imagine, Act via Holistic Language and Vision Planning for Manipulation

**Fei Ni**[1]    **Jianye Hao**[1,2*]    **Shiguang Wu**[2]    **Longxin Kou**[1]    **Yifu Yuan**[1]    **Zibin Dong**[1]
**Jinyi Liu**[1]    **Mingzhi Li**[1]    **Yuzheng Zhuang**[2]    **Yan Zheng**[1*]
[1]College of Intelligence and Computing, Tianjin University    [2]Huawei Noah's Ark Lab

## Abstract

Long-horizon manipulation tasks with general instructions often implicitly encapsulate multiple sub-tasks, posing significant challenges in instruction following. While language planning is a common approach to decompose general instructions into stepwise sub-instructions, text-only guidance may lack expressiveness and lead to potential ambiguity. Considering that humans often imagine and visualize sub-instructions reasoning out before acting, the imagined subgoal images can provide more intuitive guidance and enhance the reliability of decomposition. Inspired by this, we propose **PERIA**(**PErceive, Reason, Imagine, Act**), a novel framework that integrates holistic language planning and vision planning for long-horizon manipulation tasks with complex instructions, leveraging both logical and intuitive aspects of task decomposition. Specifically, we first perform a lightweight multi-modal alignment on the encoding side to empower the MLLM to perceive visual details and language instructions. The MLLM is then jointly instruction-tuned with a pretrained image-editing model to unlock capabilities of simultaneous reasoning of language instructions and generation of imagined subgoals. Furthermore, we introduce a consistency alignment loss to encourage coherent subgoal images and align with their corresponding instructions, mitigating potential hallucinations and semantic conflicts between the two planning manners. Comprehensive evaluations across three task domains demonstrate that PERIA, benefiting from holistic language and vision planning, significantly outperforms competitive baselines in both instruction following accuracy and task success rate on complex manipulation tasks. The details and visualizations are available at the homepage.

## 1 Introduction

Recent advances in vision-language models (VLMs), such as BLIP [1] and LIV [2], enable open-vocabulary visual recognition and multi-modal alignment, showing promise in robotic manipulation tasks specified human-provided language instructions [3, 4, 5, 6]. For semantically clear and concise instructions, such as "pick the red block on the green one", robotic agents can easily understand and complete the task in a single step using action primitives. However, when instructions become more general and complex, such as "stack the blocks as a pyramid and each layer in one color", the manipulation task can span long horizons and implicitly encapsulate multiple sub-tasks separated by action primitives, posing a major obstacle in instruction following. Current approaches often resort to decomposing complex instructions into manageable subtasks, either through language planning or vision planning based on the decomposed modality. Language planning, the more common approach, decomposes into progressive stepwise sub-instructions, which can be either predefined skill libraries

---

*Corresponding authors: jianye.hao@tju.edu.cn, yanzheng@tju.edu.cn

38th Conference on Neural Information Processing Systems (NeurIPS 2024).

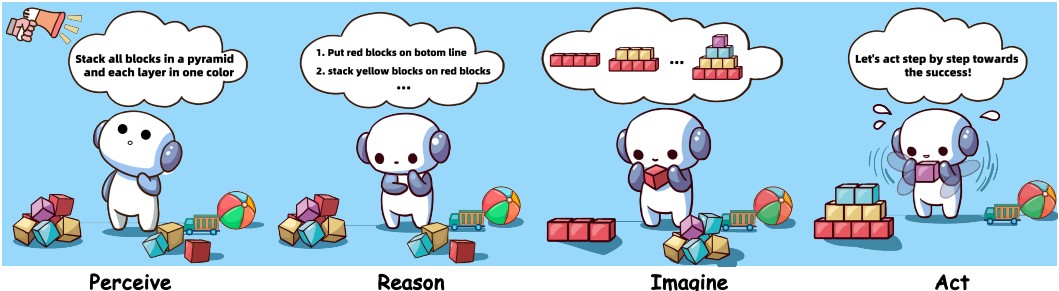

Figure 1: Overview of PERIA (Perceive, Reason, Imagine, Act), inspired by the human cognitive process of following complex instructions, which involves perceiving environment and tasks, reasoning the required language plans, and imagining the intermediate subgoal images before acting.

in natural language [3, 7] or latent codebooks [8]. On the other hand, vision planning, a more recent development, decomposes complex instructions into coherent subgoal images as keyframes [9, 10], serving as visual milestones to provide more intuitive and expressive guidance for action execution.

Language planning focuses on ***"how to act"*** and the sub-instructions outline the necessary procedural action process of the task completion, emphasizing the temporal dependencies and causal relationships between decomposed stepwise sub-instructions. On the other hand, vision planning concentrates on ***"what to act towards"*** and intuitive and grounded subgoal images with rich spatial and contextual information can enable robot agents to more easily understand what intermediate landmarks and visual anchors should achieve towards task completion. From a cognitive perspective, humans rely on a symbiotic operation of the brain's hemispheres [11], with the left primarily associated with logical ***language-based reasoning***, and the right is linked to intuitive ***visual-based imagining***. For humans, language planning and vision planning are often intertwined and performed simultaneously, involving either imagining the desired intermediate goals and then reasoning about the required plans to achieve them, or first reasoning out necessary stepwise plans and then imagining corresponding resulting images. Inspired by this, a natural question arises: *Can we develop a framework that emulates this cognitive synergy by simultaneously performing language planning and vision planning for robotic manipulation tasks involving complex instructions just like humans?*

For this, we propose **PERIA**(**PErceive, Reason, Imagine, Act**), a novel framework that integrates multi-modal large language model (MLLM) and diffusion model to enable language-based reasoning and visual-based imagining respectively, leveraging holistic language planning and vision planning for long-horizon manipulation tasks with general complex instructions. Specifically, we first train the MLLM's perception ability by fine-tuning the encoder side's projection layer to align the text and vision modalities in the LLM's hidden layers in a lightweight manner, avoid the potential hallucinations and enhance the grounding ability. Next, we perform instruction tuning to simultaneously equip PERIA reasoning and imagination capabilities by explicitly adding additional image tokens after the reasoning phase and extracting rich latent image representations from MLLM to guide the generation of corresponding subgoal images. Moreover, We also introduce an alignment loss between reasoned sub-instructions and imagined subgoal images to enhance the consistency and accuracy of vision and language planning, jointly updated with generation and reasoning losses. In this way, vision planning provides a visualization of language planning, offering more intuitive guidance to avoid potential confusion. Language planning, in turn, provides reliable logical guidance at the semantic level for vision planning, preventing semantic conflicts in the generation of coherent image chains. The comprehensive evaluation across three typical long-horizon manipulation tasks demonstrates that PERIA enjoy the accuracy of instruction following and synergistic combination significantly improves decomposition accuracy and task success rate compared to existing methods that rely solely on either language or vision planning alone. The contributions of this work are as follows:

- We propose PERIA, a novel framework that integrates holistic language planning and vision planning, leveraging the logical and intuitive decomposition of general complex instructions.
- We encourage MLLM to output rich latent visual tokens to guide the diffusion model to generate images and further explicitly align between language instructions and visual subgoals, simultaneously developing the MLLM's reasoning and the diffusion model's imagination capabilities.

- PERIA demonstrates significant improvements in instruction following and task success rate on complex manipulation tasks compared to existing methods that rely on either language or vision planning alone, establishing a promising and inspiring paradigm for long-horizon manipulation.

## 2 Related Work

### 2.1 Hierarchical Planning for Long-horizon Manipulation

Embodied manipulation tasks with general instructions often span multiple subtasks and long horizons, making direct end-to-end action prediction challenging due to compounding errors without intermediate guidance [12, 13, 14, 15]. Recent works adopt hierarchical planning, decomposing complex instructions into sequential sub-tasks to execute. Language planning like LISA [8] and Xskill [16] decompose the general instruction based on the latent skill codebook discovered during training. SayCan [3] and EmbodiedGPT [7] both leverage LLM to enable reasoning into sequential interpretable instructions in natural languages. Vision planning, a more recent development, decomposes complex instructions into sequential subgoal images. CoTDiffusion [10] utilizes diffusion models to translate multi-modal prompts into coherent subgoal images in a chain-of-thought manner, serving as visual milestones that are challenging to describe using language alone. While existing works rely solely on either language or vision planning, our PERIA framework enables simultaneous language and vision planning, harnessing the strengths of both approaches to provide a comprehensive, multi-modal guide that enhances the accuracy of instruction decomposition and following.

### 2.2 LLM for Robotics Manipulation

With the tremendous success of LLMs, there has been a surge in research exploring their capabilities for robotics manipulation, such as SayCan [3], Inner Monologue [17]. PAR [18] leverages a vision language model(VLM) as a captioner for visual observations and the generated captions are fed into LLM for language planning. ViLA [19] and CoPA [20] follow a similar pipeline but replace LLM and VLM with more advanced GPT4V [21] with stronger visual reasoning capabilities. EmbodiedGPT [7] employs a pre-trained open-sourced LLaMA model [22] as the language model for instruction tuning on collected robotics data, enhancing reasoning and planning capability specifically for embodied scenarios. PERIA introduces image generation as an additional supervision signal to encourage the MLLM to perceive more detailed visual details, reducing hallucinations and errors in reasoning. The generated images in vision planning also provide a more intuitive guide that further enhances the accuracy of instruction decomposition and improves instruction following performance.

### 2.3 Image Generation for Robotics Manipulation

Inspired by recent development of recent text-to-image models [23, 24, 25], many works have begun to explore the visualization of manipulation tasks to guide robot action execution. LfVoid [26] enables the editing of original observations to obtain goal images based on natural language instructions to provide reward signals. SuSIE [9] similarly leverages an image-editing diffusion model to act as a high-level planner by proposing intermediate subgoals that a low-level controller can accomplish. LfVoid and SuSIE are limited to single-step sub-instructions, while CoTDiffusion [10] supports various instruction modalities and generates coherent subgoal image chains using a semantic alignment module. These works demonstrate that subgoal images can provide more detailed and intuitive guidance than language-only instructions. However, they do not incorporate LLMs for reasoning and are prone to failure and semantic conflicts without logical guidance. Our PERIA framework leverages the prior knowledge in MLLMs to assist in generating promising sequential images, enhancing consistency with complex task instructions and improving instruction following.

## 3 Method

By leveraging MLLM and diffusion-based image editing models, PERIA enables holistic language planning and vision planning for stepwise language instructions and visual subgoal images, serving as language milestones and visual anchors to guide action execution in long-horizon tasks. We first introduce the lightweight alignment of language and vision modalities on the encoding side of

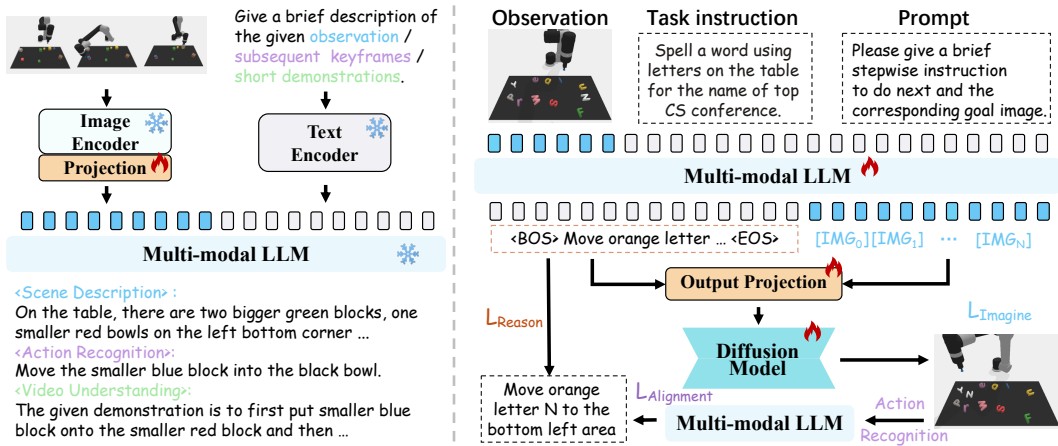

Figure 2: Overview of **PERIA**. PERIA first learns to align the vision and language on encoding side of MLLM for perceiving. Then PERIA performs instruction tuning to MLLM jointly with diffusion model in an end-to-end manner to unlock the reasoning and generation ability for holistic language planning and vision planning. 🔥 and ❄ show the module is trainable and frozen, respectively.

the MLLM to achieve precise **Perceive** ability in Section 3.1. We then illustrate how to perform instruction tuning on the MLLM to enable **Reason** for language planning in Section 3.2 and how to jointly train with a diffusion model to **Imagine** coherent subgoal images aligned with corresponding instructions in Section 3.3. Moreover, we leverage an explicit alignment between instructions and images to achieve a synergistic effect between language and vision in Section 3.4. Since our focus is not on the low-level policy, please refer to Appendix E for the implementation details of **Act**.

## 3.1 Perceive: Encoding-side LLM-centric Multimodal Alignment

To enable embodied robot agents to effectively perceive and comprehend visual scenes, a straightforward approach is to use an off-the-shelf Vision-Language Model (VLM) as an image captioner. However, the information bottleneck between the LLM and VLM limited by language modality, results in missing visual details, which is particular problematic in robotics manipulation tasks that require precise visual understanding. To address this limitation, we leverage image captioning as a training task for LLM-centric multimodal alignment to encourage visual representations compatible with the text feature space within the LLM, extending it to a MLLM to allow for a more precise and detailed comprehension of visual scenes. Specifically, we utilize the privileged information available in simulation environments to create a large-scale dataset of pairwise ground-truth data through three types of captioning tasks. First, the *single-frame scene description scenario* focuses on understanding a single observation frame, where the MLLM is tasked with providing a brief description covering aspects such as object recognition, size identification, number counting, color understanding, and spatial relationships. Second, *action recognition between consecutive keyframes scenario* presents the MLLM with two consecutive keyframes, requiring it to understand the visual difference between them and recognize the executed action, enhancing the perception of spatial relationships and action dynamics at the instruction level. Moreover, *short demonstration understanding scenario* involves processing a given short demonstration of frame sequences, strengthening the MLLM's temporal relationship understanding and grounding ability. These carefully designed captioning tasks with the high-quality training data, enable the MLLM to develop a strong foundation in visual perception and understanding for embodied manipulation tasks.

Specifically, initialized from a pre-trained LLM, the MLLM contains a visual encoder $\mathcal{V}$(*e.g.,* CLIP-L [27]) to extract the visual features $f$, and an projection layer $\mathcal{W}$ to project $f$ into the language modality. We follow the training of LLaVA [28] with cross-entropy loss (CELoss) as:

$$
\begin{aligned}
\mathcal{C} &= \{x_1, x_2, ..., x_l\}, \quad \mathcal{I} = \{v_1, v_2, ..., v_n\}, \\
\hat{x}_t &= \text{MLLM}(\{x_1, ...x_{t-1}\}|[\texttt{prompt}, \mathcal{W}(f = \{\mathcal{V}(v_i)\}_{i=1}^n)]), \\
\mathcal{L}_{\text{Perceive}} &= \sum_{t=1}^{l} \text{CELoss}(\hat{x}_t, x_t)
\end{aligned}
\tag{1}
$$

where $\mathcal{C}$ can be the image caption for features alignment and $l$ is number of word tokens. $n$ is the numbers of the images $\mathcal{I}$ fed in MLLM, which can be differ in different captioning tasks. To perform the lightweight alignment, we freeze the weights of both the vision encoder $\mathcal{V}$ and LLM, and only update the parameters of $\mathcal{W}$ that encourage to map the image features into a shared latent space that is compatible with the MLLM's hidden representations. The alignment of visual and language modalities on the encoding side can effectively alleviates hallucinations and lays the foundational perception abilities for generating more grounded language and vision planning. For more detailed categorical analysis of improvement benefiting from captioning task, please refer to Appendix F.1.

## 3.2 Reason: Instruction Tuning for Language Planning

With the initial coarse alignment of visual and language on the encoding side, we proceed to instruction tuning to encourage the MLLM to learn how to decompose complex instructions for language planning. These general task instructions $\mathcal{T}$ can be categorized into two types based on the modalities involved: 1) text-only instructions, such as "sort blocks into bowls according to the matching colors" which can be directly processed by the language encoder; and 2) multi-modal instructions that consist of interleaved language and images of a single object or whole observation, as suggested by VIMA-BENCH [13], which are more expressive and challenging to understand. For instance, consider an interleaved multi-modal prompt such as "Stack objects  in this order  ", where  serves as a placeholder for the corresponding images, which can be images of blocks or other objects in the observation. With the benefit of encoding-side alignment via several captioning scenarios across frames and videos, MLLM equipped with the input projection layer can handle multi-modal instructions including interleaved text, images, and even video frames.

Then we design an instruction prompt $\mathcal{P}$ template such as: "Given the current observation  and the general task instruction [$\mathcal{T}$], can you provide a brief and concise sub-instruction about how to act next?". We collect the stepwise language instructions $\mathcal{E}$ as the groundtruth response for the language planning task specified with the observation $o$, the prompt $\mathcal{P}$ and the general instruction $\mathcal{T}$. The instruction tuning loss for language planning is defined as follows:

$$
\begin{aligned}
\mathcal{E} &= \{e_1, e_2, ..., e_l\}, \quad \mathcal{I} = \{o, v_1, v_2, ..., v_n\}, \\
e'_t &= \text{MLLM}(\{e_1, ..., e_{t-1}\} \,|\, [\mathcal{P}, \mathcal{T}, \mathcal{W}(f = \mathcal{V}(\mathcal{I}))]), \\
\mathcal{L}_{\text{Reason}} &= \sum\nolimits_{t=1}^{l} \text{CELoss}(e'_t, e_t)
\end{aligned}
\tag{2}
$$

where $e$ are the word token of stepwise instruction, and $v$ are the possible $n$ images from multi-modal instruction. The text instruction with the  token and the corresponding image are processed by the aligned language encoder and image encoder respectively and then are unified fed into LLM for reasoning. The MLLM follows the standard auto-regressive training for the next token prediction and then can be regarded as a visual assistant for various tasks such as visual question answering. To perform instruction tuning, we fine-tune the MLLM using the LoRA technique [29] while keeping the encoding side frozen, including the visual encoder and its projection layer. Additionally, we employ two kinds of prompts to require MLLM to generate the next sub-instruction for the single step or all the stepwise instructions in order respectively. Two modes can be randomly switched during the instruction tuning and can effectively encourage MLLM to perform single-step and multi-step sequential language planning for closed-looped and open-looped control respectively.

## 3.3 Imagine: Decoding-side Synergistic Training for Vision Planning

Considering the phrase *a picture is worth a thousand words*, subgoal images could provide higher expressive capabilities for conveying subtasks compared to sub-instructions with complex language only. Inspired by this, we integrated pre-trained conditional diffusion models to convert decomposed sub-instructions into coherent visual subgoal plans. While a natural approach would be to directly use the text instructions or captions as prompts for the image editing model, shown in Figure 3, relying solely on decoded text instructions as conditions may lead to an information bottleneck. The expressiveness of the instructions can be limited, and information loss may occur, as it is confined to the language modality. Inspired by [30, 31], to bridge the gap between the language and vision modalities, we introduce $N$ special [IMG] tokens in the vocabulary codebook of the MLLM. These special tokens have trainable word embeddings and should be predicted after the generated language instructions jointly during the reasoning phase, shown in Figure 2. These appended visual tokens

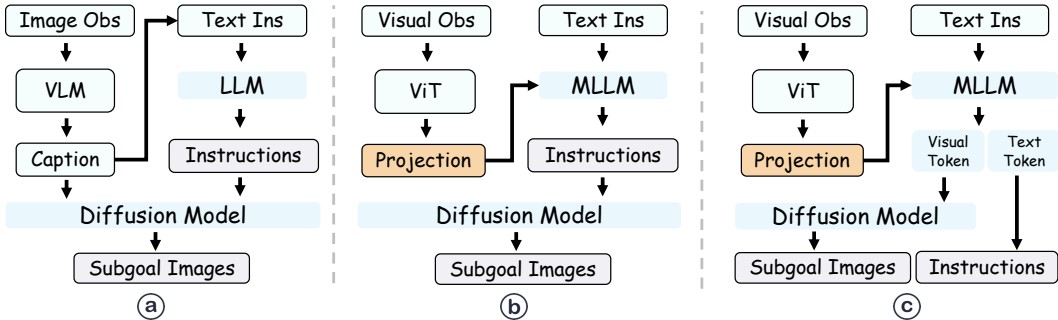

Figure 3: Three pipelines of MLLM for generation images. PERIA (ⓒ) leverage visual tokens extracted from the MLLM during language planning serve as more expressive guidance for subgoal imagination compared to captions (ⓐ) or decomposed instructions (ⓑ) in language only.

`[IMG]` are treated as latent imagination of subgoal image from the MLLM and we employ an output image projection module $\mathcal{R}$ to transform them into actual visual guidance $\mathcal{U}$ for diffusion model:

$$\mathcal{U} = \mathcal{R}(\{w_{\texttt{lang}} + h_{\texttt{[IMG]}}\}, q) \tag{3}$$

where $w$ is the word embedding of language instructions and $h$ is the hidden state from the last layer of MLLM before image projection layer of `[IMG]`, conditioned on learnable query embeddings $q = \{q_1, ..., q_L\}$, where $L$ is the token numbers setting from the pre-trained diffusion model. The transformation over $w$ can be seen as a general representation from language modality, while $h$ represents a more grounded visual imagination that aligns with the language planning within the MLLM's reasoning. To simultaneously fine-tune the diffusion model and the MLLM, we employ the generation loss between the generated image and the groundtruth image. Our image editing model is based on latent diffusion, which learns the noise latent $z_t$ at the denoising timestamp $t$ to reconstruct the groundtruth goal image. The generation loss is to learn the UNet $\epsilon_\theta$ that predicts the added noise based on the input image $v$ and the visual imagination guidance $\mathcal{U}$ from the MLLM, formulated as:

$$\mathcal{L}_{\text{Imagine}} = \mathbb{E}_{o,v,\mathcal{U},\epsilon \sim \mathcal{N}(0,1),t} \left[ ||\epsilon - \epsilon_\theta(z_t, t, v, \mathcal{U})||_2^2 \right] \tag{4}$$

### 3.4 Enhancing Consistency between Vision and Language Planning

To further enhance the consistency between vision and language planning, we introduce an additional alignment objective between generated language instructions and visual images, as illustrated in Figure 2. Specifically, we feed both the generated image $v_{t+1}$ and the current observation $o_t$ at planning step $t$ into the MLLM and prompt it with understanding the differences between the two frames, which is exactly the *action recognition* captioning task in the perceive phase of PERIA Section 3.1. The response output $\tilde{\mathcal{E}}_t$ generated by the MLLM is compared with the groundtruth stepwise language instruction $\mathcal{E}_t$ for consistency, and can be formulated as the alignment consistency loss:

$$\mathcal{C} = \{\mathcal{E}_t\}_{t=0}^T, \quad \mathcal{I} = \{(o_t, v_{t+1})\}_{t=0}^T,$$
$$\tilde{\mathcal{E}}_t = \text{MLLM}(\texttt{prompt}, \mathcal{W}(f = \{\mathcal{V}(o_t), \mathcal{V}(v_{t+1})\})), \tag{5}$$
$$\mathcal{L}_{\text{Consistency}} = \sum_{t=0}^T \text{CELoss}(\tilde{\mathcal{E}}_t, \mathcal{E}_t)$$

The additional alignment task reinforces the synergy between vision and language planning, ensuring that generated subgoal images and text instructions are consistent and mutually informative, alleviating the compounding errors that may arise in long-horizon tasks due to inconsistencies. Vision planning provides a visualization of language planning, offering more intuitive guidance and reducing potential confusion or ambiguity. Conversely, language planning provides logical guidance at the semantic level for vision planning, preventing semantic conflicts during the generation of coherent image chains. This synergistic approach leverages the complementary strengths of vision and language, enabling PERIA to produce plans that are both visually grounded and semantically meaningful.

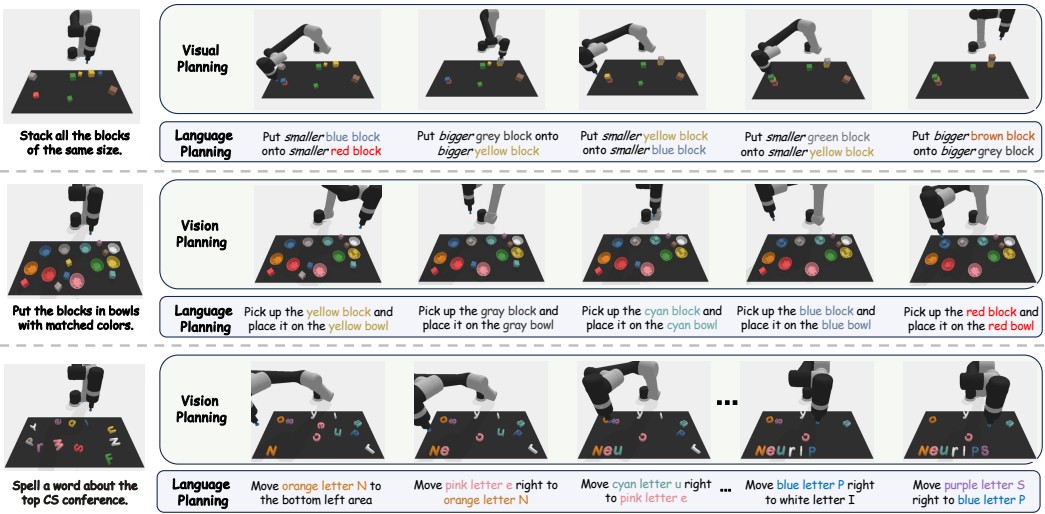

Figure 4: The illustrating examples of holistic language and vision planning for general instructions, with stepwise sub-instructions and coherent subgoal images enhancing the instruction following.

## 4 Experiments

### 4.1 Experiment Setup

**Benchmark & Tasks**    To provide comprehensive evaluations, we conduct experiments across three typical long-horizon manipulation environments. More benchmark and task details are in Appendix A.

- **LoHoRavens** [18]: is a Ravens-based benchmark consisting of 11 long-horizon language-conditioned tasks categorized into *Stacks*, *Sort*, and *Matching*. Original tasks all involve manipulating **Bowls&Blocks** and we additionally develop a more complex **Letters** scenario including 9 tasks of *Shape*, *Orders*, and *Spell* to further diversify the instruction and increase task difficulties.
- **VIMA-BENCH** [13], a benchmark for long-horizon manipulation, contains diverse tasks guided by multi-modal prompts. We choose 8 tasks from three representative categories - *Rearrange*, *Constraints*, and *Follows*, specified by interleaved language and images of object or ultimate goal.

**Baselines**    To more clearly and comprehensively evaluate the effectiveness of different approaches, we categorize the baselines into three types based on their specific planning methods as follows:

- *End-to-end*:  We choose **CLIPort** [12], one of the most widely used end-to-end language-conditioned imitation learning framework in Ravens-like manipulation benchmarks. CLIPort directly take the high-level language instructions as input to predict the action without a planner.
- *Language Planning*: We select several representative language planning methods that decompose general high-level instructions into stepwise instructions. **LISA** [8] trains a skill predictor to combine the discovered implicit skill codebooks for complex instructions. **PAR** [18] (Planner-Actor-Reporter) replaces the latent skill planner with an LLM, using the VLM as a reporter for visual observations. The instruction and the generated captions are then fed into the LLM for language planning. **EmbodiedGPT** [7] follows a similar pipeline but replaces LLM and VLM with more advanced MLLM with stronger visual reasoning capabilities after instruction tuning.
- *Vision Planning*: **SuSIE** [9] incorporating a pretrained image-editing models to generate goal images for action prediction but only support simple single-step instructions. **CoTDiffuison** [10] leverage a semantic alignment module within the diffusion model to enable the sequential subgoal image generation for complex general instructions. For more details, please refer to Appendix C.

### 4.2 Main Quantitative Results of Success Rate

We begin by comparing the performance of PERIA and baselines in solving long-horizon tasks across three typical task domains. The baselines can be categorized into three types of planners: *e2e planner*, *language planner*, and *visual planner*. As shown in Table 1, PERIA significantly outperforms other baselines in terms of success rate. As expected, the end-to-end learning method performs the worst due to the lack of intermediate guidance, making it difficult for the policy to follow general instructions for long-horizon tasks. In contrast, the *visual planner* paradigm, which explicitly

Table 1: The evaluation of success rate between baselines and we report the mean and variance across 5 seeds.

| Model | Blocks&Bowls | | | Letters | | | VIMA-BENCH | | |
|---|---|---|---|---|---|---|---|---|---|
| | Stacking | Sort | Matching | Shape | Orders | Spell | Rearrange | Follow | Constraint |
| CLIPort | $18.4_{\pm 3.2}$ | $19.2_{\pm 4.6}$ | $17.8_{\pm 2.9}$ | $9.8_{\pm 1.4}$ | $8.1_{\pm 2.7}$ | $2.3_{\pm 0.8}$ | $5.8_{\pm 1.9}$ | $2.4_{\pm 0.6}$ | $8.3_{\pm 2.1}$ |
| LISA | $26.6_{\pm 4.8}$ | $22.1_{\pm 3.5}$ | $23.0_{\pm 5.1}$ | $18.4_{\pm 2.6}$ | $16.1_{\pm 3.9}$ | $10.2_{\pm 1.7}$ | $8.9_{\pm 2.3}$ | $6.3_{\pm 1.5}$ | $11.9_{\pm 4.2}$ |
| PAR | $34.7_{\pm 5.5}$ | $32.8_{\pm 6.3}$ | $31.1_{\pm 4.4}$ | $31.5_{\pm 5.8}$ | $30.7_{\pm 4.9}$ | $27.3_{\pm 7.2}$ | $24.4_{\pm 6.1}$ | $16.1_{\pm 3.7}$ | $26.5_{\pm 4.6}$ |
| EmbodiedGPT | $48.6_{\pm 6.7}$ | $49.1_{\pm 5.9}$ | $43.4_{\pm 7.8}$ | $40.9_{\pm 6.4}$ | $48.2_{\pm 7.5}$ | $52.7_{\pm 6.2}$ | $38.3_{\pm 5.3}$ | $37.2_{\pm 4.7}$ | $43.5_{\pm 6.9}$ |
| SuSIE | $34.1_{\pm 3.8}$ | $32.6_{\pm 4.1}$ | $33.2_{\pm 5.7}$ | $37.8_{\pm 6.6}$ | $35.2_{\pm 4.3}$ | $34.1_{\pm 7.4}$ | $37.9_{\pm 6.8}$ | $40.2_{\pm 5.4}$ | $51.2_{\pm 7.1}$ |
| CoTDiffusion | $47.9_{\pm 6.0}$ | $44.3_{\pm 7.6}$ | $56.6_{\pm 5.2}$ | $46.1_{\pm 6.5}$ | $53.9_{\pm 4.8}$ | $44.8_{\pm 7.9}$ | $51.2_{\pm 6.3}$ | $54.5_{\pm 7.3}$ | $76.1_{\pm 5.6}$ |
| PERIA (ours) | $\mathbf{63.9}_{\pm 5.8}$ | $\mathbf{65.0}_{\pm 6.4}$ | $\mathbf{72.3}_{\pm 7.1}$ | $\mathbf{60.6}_{\pm 5.2}$ | $\mathbf{65.2}_{\pm 6.7}$ | $\mathbf{71.1}_{\pm 7.5}$ | $\mathbf{74.8}_{\pm 6.0}$ | $\mathbf{67.2}_{\pm 7.8}$ | $\mathbf{89.3}_{\pm 4.9}$ |

decomposes tasks into stepwise instructions and employs a hierarchical framework consisting of a language planner and a language-conditioned policy, shows more promise and demonstrates a clear advantage over the end-to-end approach. Within the *visual planner* category, PAR and EmbodiedGPT both leverage the common sense knowledge from LLM and significantly outperform LISA, which uses a skill predictor for latent skill codebook rather than LLM. Furthermore, although both PAR and EmbodiedGPT are based on LLaMA, EmbodiedGPT employs a visual projector to expand the LLM to an MLLM for more precise perception and reasoning capabilities, while PAR applies a captioner to convert visual images into the language modality for reasoning, which may impact the accuracy of reasoning and task success rate to some extent. The *visual planner* paradigm, which generates intermediate keyframes, offers more intuitive guidance compared to language planning, and its advantage is more evident in VIMA-BENCH, where sub-tasks are challenging to describe sufficiently using language-only instructions. CoTDiffusion supports generating coherent subgoal images for complex instructions, resulting in performance gains compared to SuSIE. But CoTDiffusion does not explicitly reason about the instructions, which can lead to semantic inconsistencies in the generated subgoal images, causing it to still underperform compared to our algorithm. In contrast, our PERIA algorithm introduces an MLLM for explicit reasoning and generation, providing more sufficient and reliable intermediate guidance for instruction following in long-horizon tasks.

## 4.3 Further Analysis

**Accuracy of Language Planning** We compare the accuracy of language planning with two evaluation metrics: the token accuracy which directly calculates the token-level matching rate between decomposed stepwise instructions and the groundtruth instructions, and the semantic similarity by calculating the embeddings distance of two instructions from pre-trained text encoder like CLIP. Our focus here is on generative language planning using LLMs and exclude LISA from this comparison.

Table 2: Evaluation of reasoning accuracy between methods on two metrics.

| Method | Token ↑ | Semantic ↑ |
|---|---|---|
| PAR | 58.2 | 0.63 |
| EmbodiedGPT | 65.9 | 0.68 |
| PERIA (ours) | **97.6** | **0.98** |
| - w/o perceive pretrain | 80.2 | 0.83 |
| - w/o vision planning | 83.7 | 0.79 |

As illustrated in Table 2, PERIA demonstrates the highest accuracy in both token-level and semantic-level comparisons. Although PAR introduces LLMs for language planning, it relies on an isolated, out-of-the-shell VLM as a captioner to convert visual observations into language descriptions, which may cause details missing during hard captioning. EmbodiedGPT further introduces a projection layer to bridge the gap between vision and language in the latent space, gaining more advantages in perception which is critical in language planning. Compared to EmbodiedGPT, PERIA's superior performance can be attributed to the explicit incorporation of vision planning. By jointly fine-tuning MLLM using the additional image generation loss, the supervision from visual aspects encourages promoting attention to visual details and spatial information for more grounded reasoning. When we remove the joint training of vision planning, we observe the more frequent hallucinations and errors in language planning, such as generating unseen objects with wrong colors, sizes, or locations, which significantly decreases the accuracy of language planning. Moreover, we also ablate the encoding-side multimodal alignment and the degradation in accuracy highlights the importance of enhancing the foundational perception capabilities through our carefully designed dataset, which includes various perception-related data such as spatial relationships, temporal relationships, size recognition, and color identification. To further investigate the improvement in foundational perception abilities, we conduct a detailed categorical analysis, which can be found in Appendix F.1.

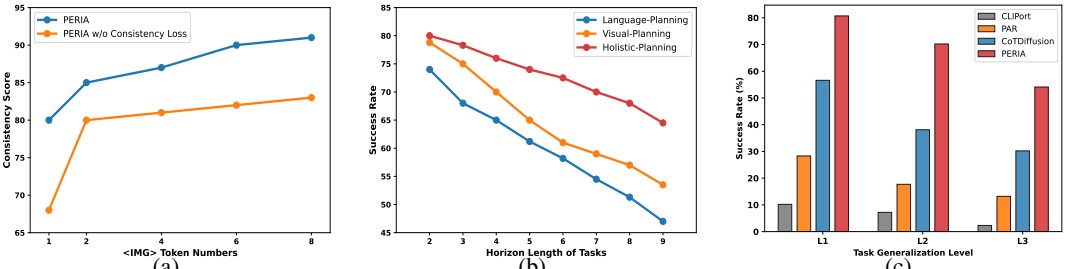

Figure 5: More detailed quantitative analysis. (a) The ablation studies on consistency loss and `[IMG]` token numbers. (b) The comparisons of three planning paradigms in tasks with various horizon lengths. (c) The evaluation of generalization ability of three levels. See text for further discussion.

**Fidelity of Vision Planning** We further compare the fidelity of generated goal images against groundtruth keyframes using the Fréchet Inception Distance (FID) [32] as the evaluation metric. Although SuSIE is not a strict vision planning method for long-horizon manipulation due to its limitation to simple single-step instructions, we grant it a relaxed privilege by providing oracle stepwise instructions to enable a comparison.

Table 3: Comparisons of FID (↓) between methods on three task domains.

| Methodology | Blocks | Letters | VIMA |
|---|---|---|---|
| SuSIE (+oracle) | 18.9 | 18.1 | 19.4 |
| CoTDiffusion | 13.1 | 15.8 | 17.6 |
| PERIA (ours) | **10.2** | **13.5** | **11.4** |
| - w/o alignment | 12.3 | 14.2 | 15.9 |

However, as shown in Table 3, PERIA still demonstrates superiority, primarily due to the implicit generation of latent image tokens during language planning. The extracted image latent embeddings from MLLM retain more details and provide more sufficient guidance beyond language for subgoal image generation. CoTDiffusion supports general instruction inputs and can sequentially generate multiple images. However, the absence of explicit language planning in CoTDiffusion makes it challenging to ensure the semantic coherence of the generated images, potentially leading to dilemmas such as semantic repetition, jumping, or regression within the generated image sequences. In contrast, PERIA incorporates MLLM for reliable instruction decomposition and leverages the extracted image latent embeddings to achieve superior fidelity in vision planning compared to existing methods. Moreover, the performance drop in the ablation study without consistency loss highlights the importance of alignment between reasoned stepwise instructions and generated subgoal images, attributed to the synergistic combination of language planning and vision planning in our framework.

**Consistency between Reasoning and Imagining** We leverage CLIP [27] to measure the image-language similarity between generated instructions and images, with results presented in Figure 5a. The additional consistency alignment loss explicitly constraints and encourages semantic alignment between the imagined images from vision planning and the reasoned stepwise instructions from language planning, significantly enhancing the collaboration and consistency between the two modalities. Furthermore, increasing the number of `[IMG]` tokens provides more expressive and sufficient guidance, facilitating the MLLM in producing semantically coherent language and image tokens. However, the benefit of adding more tokens becomes marginal beyond a certain threshold.

**Effectiveness of Holistic Planning** We modify the low-level policy model into several variants, including ones that simultaneously utilize stepwise instructions and subgoal images, as well as those that rely on each modality individually. As shown in Figure 5b, the holistic planning approach achieves the highest success rate than single planning, with the benefit of the increased amount of information available and rich multi-modal guidance for decision-making, which reduces the training difficulty of low-level policy and enhance the accuracy of action prediction. Moreover, the advantage of holistic planning becomes more evident as the horizon length increases, demonstrating its scalability and effectiveness in handling complex, long-horizon manipulation tasks.

**Generalization across Tasks** We evaluate the generalization ability in three levels with increasing difficulty: placement generalization with novel placement of objects (L1), object generalization with novel objects (L2), and combinatorial generalization with extra novel instructions (L3). The results in Figure 5c demonstrate that PERIA enjoys a substantial advantage over other baselines, highlighting the importance of the common knowledge prior within the MLLM and diffusion model and holistic planning, which enhance the generalization and robustness for unseen challenging tasks.

# 5 Conclusion

We propose **PERIA** (**SEe, Reason, Imagine, Act**), a novel framework that integrates MLLM and diffusion-based image editing models to enable holistic language and vision planning for long-horizon manipulation tasks with complex instructions. We first perform a lightweight multi-modal alignment to enhance the MLLM's fundamental perception capabilities of visual details for manipulation, alleviating potential hallucinations. Then, we encourage MLLM to output rich latent visual tokens to guide diffusion model in generating images and explicitly align language instructions with visual subgoals to simultaneously unlock MLLM's reasoning and diffusion model's imagination capabilities. Extensive evaluations across three challenging benchmarks demonstrate that PERIA significantly outperforms competitive baselines in both instruction following accuracy and task success rate, while also enjoying better generalization ability across tasks. We believe PERIA highlights the potential of holistic language and vision planning and we hope this novel paradigm can provide some insights to robotics manipulation research of long-horizon tasks with complex instructions in free-form, towards more open embodied scenarios. One current bottleneck is the relatively high time cost of training and inference. Improving the joint training efficiency of MLLMs and diffusion models in a lightweight manner and accelerating image generation sampling are interesting directions for future work.

# 6 Acknowledgements

This work is supported by the National Natural Science Foundation of China (Grant Nos. 62422605, 92370132, 62106172), the National Key R&D Program of China (Grant No. 2022ZD0116402) and the Xiaomi Young Talents Program of Xiaomi Foundation.

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

# A  Details of Benchmarks and Tasks

To provide comprehensive evaluations, we conduct experiments across three typical long-horizon manipulation environments covering diverse instruction types and task types.

## A.1  Bowls&Blocks

LoHoRavens [18] is a benchmark dataset built upon the Ravens robot simulator, comprising ten long-horizon, language-conditioned tasks. The tasks are categorized into three types: *Stacks*, *Sort*, and *Matching*. In *Stacks* tasks, the objective is to place blocks in absolute or relative areas. *Sort* tasks require sorting blocks or bowls with similar specified attributes together. *Matching* tasks involve placing corresponding blocks into matching bowls. These tasks encompass various aspects of long-horizon reasoning, including color, size, space, arithmetic, and reference. To successfully complete each task, the robot must effectively combine multiple reasoning capabilities and develop an appropriate long-horizon plan.

**Move**  Move the blocks with some specified attributes like colors, sizes or locations to the specified area. We design 4 tasks as follows:

- **MoveBlocktoArea:** `Move all the blocks to the {abs_area}.`
- **MoveColorBlocktoArea:** `Move all the {color} blocks to the {abs_area}.`
- **MoveBlockinAreatoArea:** `Move all blocks in {abs_area} to {abs_area}.`
- **MoveSizeBlocktoCorner:** `Move all {size} blocks to {position} corner.`

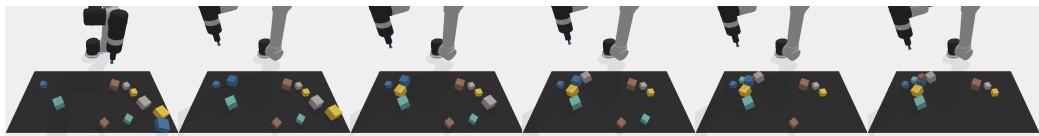

Move all blocks in right bottom area to the left top area.

Figure 6: The example in **MoveBlockinAreatoArea** in **Blocks&Bowls Move**.

**Stack**  Stack blocks or bowls with specified attributes and put to some area together. We design 4 tasks as follows:

- **StackAllBlocks:** `Stack all the blocks together.`
- **StackBlocksOfSameSize:** `Stack all the blocks of the same size.`
- **StackBlocksOfSameColor:** `Stack all the blocks of the same color.`
- **StackColorBlockstoArea:** `Stack all blocks of primary color on left side.`

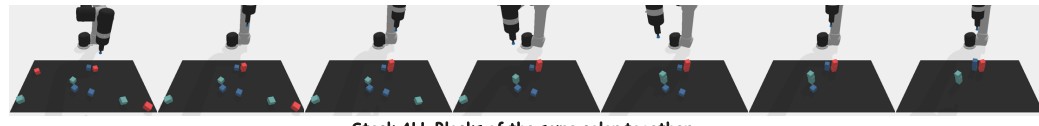

Stack ALL Blocks of the same color together.

Figure 7: The example in **StackBlocksOfSameColor** in **Blocks&Bowls Stack**.

**Matching**  Placing corresponding blocks into matching bowls or zones. We design 3 tasks as follows:

- **PutBlockInMatchingBowl:** `Stack all the blocks together.`
- **PutBlockInMismatchingBowl:** `Stack all the blocks of the same size.`
- **PutBlockinZonewithMatchingColor:** `Put blocks of the same color in the zone with matching color.`

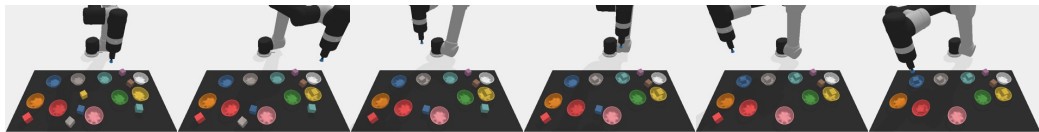

Put the blocks in the bowls with matched colors

Figure 8: The example in **PutBlockInMatchingBowl** in **Blocks&Bowls Matching**.

## A.2 Letters

In addition to the original LoHoRavens simulator, which consists of long-horizon tasks involving only **Bowls&Blocks**, we have developed a novel **Letter** scenario to further diversify the range of long-horizon reasoning tasks. This scenario randomly generates various letters with different colors and cases. We have designed three new task types: *Shapes*, which select the letters with specified shapes and arrange it it together; *Orders*, which involves arranging letters in a specific order to test the robot's understanding of sequence and position; and *Spell*, which assesses the robot's capacity for letter combination and word spelling by requiring the robot to spell words that meet specific requirements.

**Shape**   Move the blocks with some specified attributes like colors, sizes or locations to the specified area. We design 3 tasks as follows:

- **SortVerticalSymmBlockstoArea:** `Sort the vertically symmetrical letters to the bottom side.`
- **SortHorizontalSymmBlockstoArea:** `Sort the horizontal symmetrical letters to the blank space .`
- **SortCentralSymmBlockstoArea:** `Sort the central symmetrical letters to the corner.`

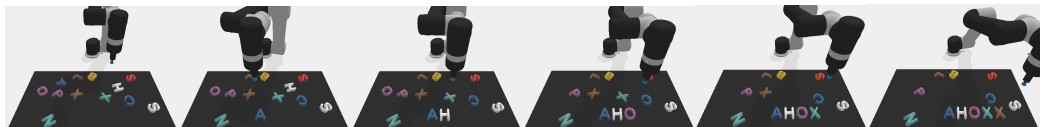

Sort the vertically symmetrical letters to the bottom side.

Figure 9: The example in **SortVerticalSymmBlockstoArea** in **Letters Shape**.

**Orders**   Stack blocks or bowls with specified attributes and put to some area together. We design 3 tasks as follows:

- **PutLettersAlphabeticalOrder:** `Put the letters on the tables in alphabetical order.`
- **PutLettersRevAlphabeticalOrder:** `Put the letters on the tables in reverse alphabetical order.`
- **SortConsLettersOrder:** `Sort the consonants from all letters in orders.`

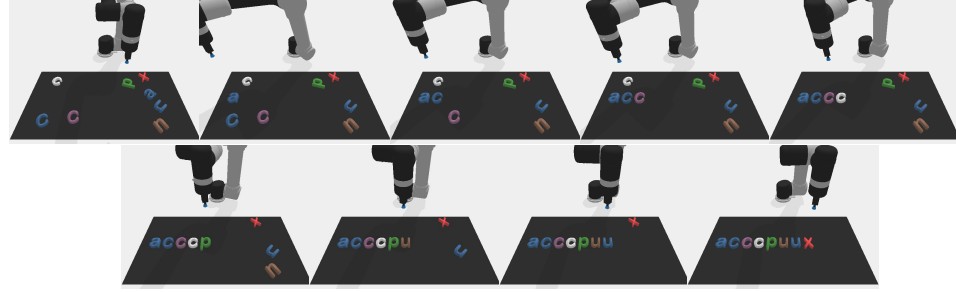

Sort all letters in alphabetical order.

Figure 10: The example in **PutLettersAlphabeticalOrder** in **Letters Orders**.

**Spell**   Placing corresponding blocks into matching bowls or zones. We design 3 tasks as follows:

- **SpellLongWords:** `Spell words that are as long as possible.`
- **SpellCSConfName:** `Spell out the name of a top CS conference.`
- **SpellTransName:** `Spell out the name of a common transportation.`

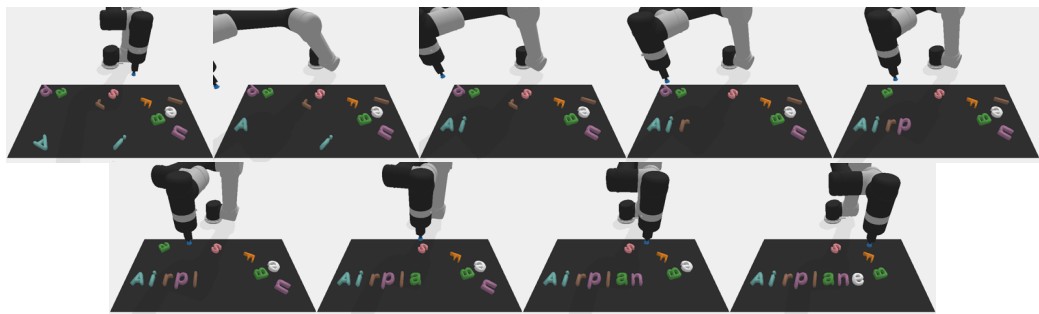

Spell a word about the transportation.

Figure 11: The example in **SpellTransName** in **Letters Spell**.

### A.3   VIMA-BENCH

VIMA-BENCH [13]: a benchmark for long-horizon manipulation with general instruction specified by multi-modal prompts, containing various tasks ranging from simple object manipulation to multi-object manipulation. We select three kinds of representative long-horizon manipulation tasks - *Rearrange*, *Constraints*, and *Follows*, in which general instructions are specified by interleaved language and images of object or ultimate goal.

**Rearrange**   Move the blocks with some specified attributes like colors, sizes or locations to the specified area. We design 2 tasks as follows:

- **RearrangeObjtoGoal:** `Rearrange objects to this setup .`
- **RearrangeObjtoGoalthenRestore:** `Rearrange objects to this setup  then restore.`

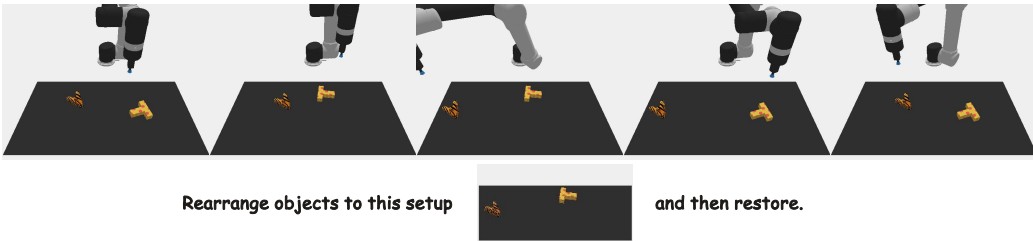

Rearrange objects to this setup            and then restore.

Figure 12: The example in **RearrangeObjtoGoalthenRestore** in **VIMA-BENCH Rearrange**.

**Constraints**   Stack blocks or bowls with specified attributes and put to some area together. We design 4 tasks as follows:

- **SweepNoExceedCons:** `Sweep all <obj> into <container> without exceeding <constraint>.`
- **SweepNoTouchCons:** `Sweep all <obj> into <container> without touching <constraint>.`
- **PutSameTextfromGoal:** `Put all objects with same texture as  into it .`
- **PutSameShapefromGoal:** `Put all objects with same shape as  into it.`

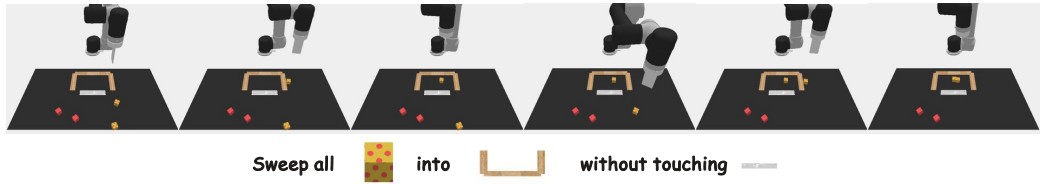

Figure 13: The example in **SweepNoTouchCons** in **VIMA-BENCH Constraints**.

**Follow**    Placing corresponding objects following orders specified by several relevant images. We design 2 tasks as follows:

- **FollowMotionObj:**    `Follow this motion for <obj>:   ...  `.
- **StackObjFollow:** `Stack objects in this order:   ...  `.

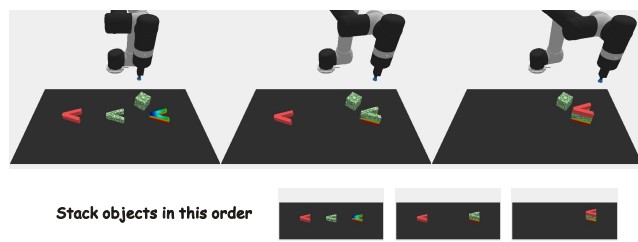

Figure 14: The example in **StackObjFollow** in **VIMA-BENCH Follow**.

## B    Details of Datasets

### B.1    The Collection of Expert Demonstrations

For the LoHoRavens and VIMA-Bench datasets, we utilize the provided oracle engines to collect expert demonstrations. It is worth noting that if there are multiple correct answers or multiple orders to complete the task, we only focus on whether the instruction-specified complex task is completed in the end and include all correct demonstrations as training data. Across all designed 28 tasks, we collect 2k demonstrations per task, gathering a total of 56k long-horizon demonstrations with horizons ranging from 2 to 10+ sub-tasks. For all the collected data, we divide it into an 80% proportion for the training dataset $\mathcal{D}^{train}$ and 20% for the testing dataset. Among all the tasks, the Spell task in the Letters dataset requires additional explanation. Unlike other tasks where the instructions are relatively fixed, the Spell task's instructions are more flexible and diverse. For example, if the instruction asks to spell the name of a top computer science conference, we can solve this by maintaining a list of all top CS conferences in advance and checking all possible permutations of the given letters to find the answer, which is then executed by the oracle engine to render the expert data. However, for more diverse instructions, such as spelling the name of a food or city, the list may be extremely large, requiring the introduction of LLMs like GPT to assist in finding the answer for the oracle engine, which is a good direction to expand the Letters domain and enrich the task types, as future work.

### B.2    The Privileged Information Annotation of Datasets

During the initialization process, we maintain a record of the assets used and annotate their corresponding attributes, enabling accurate identification of the color, size, and spatial relationships of the manipulated objects during each subtask's pick-and-place operation without the need for additional manual annotations or reliance on VLMs for captioning, which often have low accuracy without fine-tuning, even for models like GPT-4V. This privileged information from the simulator's underlying data allows us to construct a series of captioning tasks that help improve PERIA's foundational capabilities in visual perception and reasoning. During testing, access to the underlying environment information, such as the exact number of ground truth blocks or letters and their various attributes, is

not possible. The model must directly perceive and ground these crucial visual details from the visual observations and perform subsequent reasoning, significantly increasing the task's difficulty.

Table 4: Overview of three main task types, including **Block&Bowls**, **Letters** and **VIMA-BENCH**.

| Task Type | Description | Horizon | Color | Size | Spatial | Instruction |
|---|---|---|---|---|---|---|
| **Blocks & Bowls** | | | | | | |
| Move | Move all the blocks to the [ABS POS] area | 4~15 | ✗ | ✗ | ✔ | Text-only |
| | Move all blocks of a color to the red zone | 2~15 | ✗ | ✗ | ✔ | Text-only |
| | Move all the blocks in the [ABS POS] area to the [ABS POS] area | 2~15 | ✗ | ✗ | ✔ | Text-only |
| | Move all the blocks on the corner/side | 4~15 | ✗ | ✗ | ✔ | Text-only |
| Stack | Stack all the blocks | 4~15 | ✗ | ✗ | ✔ | Text-only |
| | Stack blocks of the same size. | 4~15 | ✗ | ✔ | ✔ | Text-only |
| | Stack blocks in alternate colors. | 2~15 | ✔ | ✗ | ✔ | Text-only |
| | Stack only the primary color blocks on the left side. | 2~12 | ✔ | ✗ | ✔ | Text-only |
| Matching | Put the blocks in the bowls with matching colors | 2~12 | ✔ | ✗ | ✔ | Text-only |
| | Put the blocks in the bowls with mismatching colors | 2~12 | ✔ | ✗ | ✔ | Text-only |
| | Put blocks of the same color in the zone with matching color | 2~12 | ✔ | ✗ | ✔ | Text-only |
| **Letters** | | | | | | |
| Shape | Sort the vertically symmetrical letters to the bottom side | 2~15 | ✗ | ✗ | ✔ | Text-only |
| | Sort the horizontal symmetrical letters to the blank space | 2~15 | ✗ | ✗ | ✔ | Text-only |
| | Sort the central symmetrical letters to the corner | 2~15 | ✗ | ✗ | ✔ | Text-only |
| Orders | Put the letters on the tables in alphabetical order | 2~15 | ✗ | ✔ | ✔ | Text-only |
| | Put the letters on the tables in reverse alphabetical order | 2~15 | ✗ | ✔ | ✔ | Text-only |
| | Sort the consonants from all letters in orders | 2~15 | ✗ | ✔ | ✔ | Text-only |
| Spell | Spell words that are as long as possible | 4~15 | ✔ | ✗ | ✔ | Text-only |
| | Spell out the name of a top CS conference | 4~10 | ✔ | ✗ | ✔ | Text-only |
| | Spell out the name of a common transportation | 4~15 | ✔ | ✗ | ✔ | Text-only |
| **VIMA-BENCH** | | | | | | |
| Rearrange | Rearrange the objects to this  | 2~5 | ✔ | ✔ | ✔ | Multi-modal |
| | Rearrange the objects to this  then restore | 3~10 | ✔ | ✔ | ✔ | Multi-modal |
| Constraints | Sweep all <obj> into <container> without exceeding <constraint> | 2~6 | ✔ | ✔ | ✔ | Multi-modal |
| | Sweep two <obj> into <container> without touching <constraint> | 2~9 | ✔ | ✔ | ✔ | Multi-modal |
| | Put all objects with same texture as  into it | 2~8 | ✔ | ✔ | ✔ | Multi-modal |
| | Put all objects with same shape as  into it | 2~8 | ✔ | ✔ | ✔ | Multi-modal |
| Follow | Follow this motion for <obj>: ... | 2~8 | ✔ | ✔ | ✔ | Multi-modal |
| | Stack objects in this order: ... | 2~8 | ✔ | ✔ | ✔ | Multi-modal |

## B.3 The Wordcloud of Language Instructions

To visually summarize and showcase the frequency of all instructions, including object nouns, colors, sizes, and verbs, we create a word cloud visualization in Figure 15. We tokenize each instruction and record *all* the tokens from the language instruction for each skill code used in the trajectory. Once we have this mapping from skills to tokens, we can generate heat maps and word clouds. These word distributions effectively visualize the scope of the benchmarks, which focus on manipulating objects in human spaces by following general complex instructions in unpredictable scenarios.

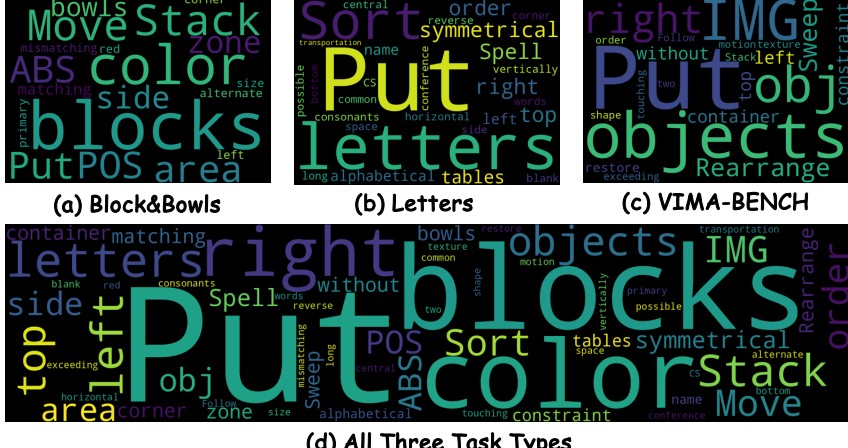

Figure 15: **World Cloud:** We created the word cloud to visually summarize the key aspects covered by the diverse manipulation instructions across three tasks types .

# C Details of Baselines

**CLIPort** CLIPort [12] is a popular end-to-end algorithm functioning as a language-conditioned imitation learning agent that directly takes in high-level language instructions without a planner. It combines the broad semantic understanding of CLIP [33] with the spatial precision of Transporter [34]. As an end-to-end baseline, we make no modifications to CLIPort, as its native SE(2) action space is well-suited for benchmarks like Ravens, which is one of the key factors contributing to the high data efficiency of Transporter and CLIPort. We train CLIPort by matching general instructions with pairwise actions for Block&Bowls and Letters. To accommodate the multi-modal instructions in VIMA-BENCH, we make additional adaptations by directly borrowing the prompt tokenization mechanism from VIMA without further modification, the same with other baselines. Specifically, instead of operating on raw RGB images, VIMA adopts an object-centric representation by cropping objects from both prompt and observation images as object tokens sequences with pixel coordinate information.

**LISA** We also compare with LISA [8], a hierarchical imitation learning framework that discovers implicit skills and learns to combine them for complex tasks. LISA learns diverse, interpretable primitive behaviors or skills from language-conditioned demonstrations to better generalize to unseen instructions. It employs vector quantization to learn discrete skill codes that are highly correlated with language instructions and the behavior of the learned policy. LISA can be considered a form of language planning, where the predicted instructions are in the form of skill codes. The low-level foundation model in LISA uses a decision transformer as its backbone and we retain the original implementation without any additional modifications.

**PAR** PAR [18] (Planner-Actor-Reporter) is a paradigm that replaces the skill predictor with an LLM, using a VLM as a reporter for visual observations. The instruction and the generated captions are then fed into the LLM for language planning. In PAR, Llama 2 13B [22] and VLM OpenFlamingo [35] with few-shot prompting are employed as the Planner and Reporter, respectively. It is important to note that the Actor, or the low-level foundation model, is precisely the language-conditioned CLIPort trained by stepwise sub-instructions, as mentioned earlier. To ensure fair comparisons, we make no modifications and keep the low-level foundation model consistent across all other baselines with CLIPort as backbones, the same with PAR.

**EmbodiedGPT** EmbodiedGPT [7] is a standard paradigm that incorporates an MLLM for language planning. The main difference between EmbodiedGPT and PAR lies in the replacement of the LLM+VLM combination with a more advanced MLLM, which possesses stronger visual reasoning capabilities. EmbodiedGPT trains the MLLM with the constructed embodied chain-of-thought dataset to enable the MLLM to perceive visual details in its hidden layers, similar to LLaVA [28]. To ensure fair comparisons, we make no modifications to the planning module and keep the low-level foundation model consistent across all other baselines, using CLIPort as the backbone, identical to the approach in PAR.

**SuSIE** SuSIE [9] proposes a hierarchical framework that leverages an image-editing diffusion model to act as a high-level planner by proposing intermediate subgoals that a low-level controller can accomplish. It is worth noting that SuSIE is not a strict vision planning method for long-horizon manipulation, as it can only support relatively simple single-step instructions and falls short when it comes to complex general instructions. To enable a comparison, we grant SuSIE a relaxed privilege by providing oracle stepwise instructions, as it is limited to handling sub-instructions of a single step and cannot generate image chains for complex general instructions. SuSIE chooses InstructPix2Pix [36] as the pre-trained image-editing model and fine-tunes it with a dataset of language-labeled video clips and robot trajectories from CALVIN [37]. Since the image editing model is sensitive to training data, we find that its generation performance on the Ravens domain is limited. To address this, we perform additional fine-tuning, keeping the number of training iterations and dataset exactly the same with PERIA.

**CoTDiffusion** CoTDiffusion [10] is a standard vision planning paradigm that supports translating general complex instructions, including text-only or multi-modal prompts, into visual subgoal images in a chain-of-thought manner. Compared to SuSIE, the most significant difference lies in CoTDiffusion's explicit design of a semantic alignment module within the diffusion model to capture the correspondence and semantic completion between the generated images and the general

instruction, enabling chain-of-thought generation. Similar to SuSIE, we fine-tune CoTDiffusion on our collected dataset and employ the same low-level image-conditioned policy as SuSIE, which is the image-conditioned variant of CLIPort. However, since CoTDiffusion does not explicitly introduce an LLM for planning, it may still encounter semantic conflicts during the vision planning process, such as repetition, backtracking, or skipping steps.

## D    Details of High-level Planner Learning

### D.1    Pretraining of Perceiving Stage

In the initial pretraining stage, PERIA aims to acquire vision-language knowledge and alignment between vision and LLM from a large collection of aligned image-text pairs. The designed captioning task for alignment of visual and language modalities in the encoding side is crucial for effective understanding and reasoning about visual scenes, bridging the gap between perception and reasoning in manipulation tasks and laying the foundation for the subsequent development of reasoning and imagination abilities in PERIA. We choose ViT-B-32 [2] as the visual encoder and Vicuna-7B [3] as our LLM backbone. For the input visual projection, we opt for a simple linear projection module, as we found that more complex architectures like Q-former from BLIP2 [1] yield similar performance. The linear projection consists of three layers with a hidden size of 4096.

We regard the output from the injected projection layer as a soft prompt for the LLM, prompting it to generate the corresponding ground-truth texts. Throughout the entire pretraining process, both the pre-trained vision encoder and the LLM remain frozen, with only the linear projection layer being fine-tuned. To perform the lightweight alignment, we freeze the weights of both the vision encoder and LLM, and only update the parameters of the projection module to encourage mapping the image features into a shared latent space that is compatible with the MLLM's hidden representations. Specifically, we train with a batch size of 64, using 8 V100 Nvidia GPUs for parallel training in 8 hours. We employ the AdamW optimizer [38] with a learning rate of 2e-4, a linear warmup of 1k steps, and a weight decay of 0.01.

### D.2    Joint Training of Reasoning and Imagining

The output projection layer adopts a transformer-based architecture characterized by a hidden size of 512, 4 attention heads, 4 encoder layers, and 4 decoder layers. The latent image token embeddings and the word embedding are fed into the output projection layer and map the latent image representation into the latent space of the diffusion model. We train the MLLM and diffusion model jointly using instruction-following datasets and incorporate LoRA [29] to fine-tune the weights of the LLM to achieve lightweight supervised fine-tuning. For the diffusion-based image editing model, we choose the finetuning pipeline borrowed from Instruct-Pix2Pix [36], the most widely used pipeline for conditional image editing tasks. We train for 50k steps with a batch size of 16, using 8 V100 Nvidia GPUs for parallel training over 42 hours. We employ the AdamW optimizer [38] with a learning rate of 1e-4, a linear warmup of 1k steps, and a weight decay of 0.01. We track an exponential moving average (EMA) of the model parameters with a decay rate of 0.999 and use the EMA parameters at test time. The strength of classifier-free guidance $\omega$ is set to 2.0, and we use the DDIM sampler [39] with 50 sampling steps.

Empirically, we find that the pretraining stage of perception is crucial. Without the encoding pretraining in the perception stage, the convergence time of the subsequent decoding side significantly increases, and the performance deteriorates. During the joint training of MLLM and the diffusion model, we observe that the loss convergence of the reasoning stage is often faster than that of the generation stage. One major reason could be that image editing is more challenging, requiring more details compared to language planning. We did not extensively tune the ratios between the image loss, generation loss, and consistency loss. Instead, we simply added them together considering the similar scaling ranges among them. Investigating the optimal weighting of these loss components could potentially further improve the synergy between language and vision planning, but we leave this exploration for future work. Furthermore, we notice that when the image loss of the diffusion model approaches convergence, continued training, although not resulting in a significant decrease in loss,

---

[2] https://huggingface.co/sentence-transformers/clip-ViT-B-32
[3] https://huggingface.co/lmsys/vicuna-7b-v1.5, 7B, version 1.5

notably improves the fidelity of the generated images during evaluation. Therefore, after reaching a certain level, we turn off the gradients of the LLM and keep only the gradients of the diffusion model enabled, which can further accelerate the convergence speed of the generation loss without affecting the overall reasoning quality. However, we have not further investigated the specific relationship, as it is not our main focus, but it presents an interesting research direction that we will explore in future work. The summary of architecture and the parameters are listed in Table 5 as follows:

Table 5: The overall configuration and training pipeline of two training phases for MLLM and diffusion model.

| Training Stage | Vision Encoder | | Input Projection | | LLM | | Output Projection | | Diffusion | |
|---|---|---|---|---|---|---|---|---|---|---|
| | Name | Param | Name | Param | Name | Param | Name | Param | Name | Param |
| Perceiving | ViT-B-32 | 87M ❄ | Linear | 18M 🔥 | Vicuna [40] | 7B ❄ | - | - | - | - |
| Reason& Imagine | ViT-B-32 | 87M ❄ | Linear | 18M ❄ | Vicuna [40] | 7B 🔥 | Transformer | 31M 🔥 | SD [41] | 1.3B 🔥 |

# E  Details of Low-level Policy Learning

In the final phase of the PERIA framework, we focus on training the low-level policy model to develop its capability to act, enabling the effective execution of the generated language and vision plans from the previous phases. We adopt CLIPort, a widely used end-to-end learning algorithm for Ravens, as its native SE(2) action space is well-suited for benchmarks like Ravens, which is one of the key factors contributing to the high data efficiency of Transporter and CLIPort. CLIPort has two variants: a language-conditioned policy and an image goal-conditioned policy. We train these variants with stepwise language sub-instructions and coherent keyframes as inputs, respectively, allowing them to serve as the low-level foundation policy models for language planning and vision planning. For a fair comparison, our planning-based methods, whether language planning or vision planning, use CLIPort as the backbone for the low-level foundation model.

To accommodate the simultaneous presence of stepwise language sub-instructions and coherent keyframes from vision planning and language planning, we design a variant that is conditioned on both image and sub-instruction simultaneously. We combine the two representations through a cross-attention block with a 4-layer lightweight attention layer, 4 cross-attention heads, and an embedding dimension of 768 as the fusion layer. We sample the oracle action trajectories $a$, current observation $o$, stepwise instruction $e$, and pair-wise subgoal images $v$ from the $\mathcal{D}^{train}$ as mini-batch $\mathcal{B}$. The action $\hat{a}$ is predicted with the instruction $e$ and corresponding $v$ simultaneously. The low-level policy $\psi$ is updated on mini-batch $\mathcal{B}$ according to the following loss:

$$\mathcal{L}_{\text{Act}} = \sum_{t=1}^{T} ||\hat{a}_t - p_\psi(a_t|o_t, e_t, x_t)|| \tag{6}$$

We use the AdamW optimizer [38] with a learning rate of 1e-4, a linear warmup of 500 steps, and a weight decay of 0.01. We train with a batch size of 64 for 10k steps on a single V100 Nvidia GPU, which takes 12 hours. The policy model is conditioned on both the generated subgoal images and the reasoned language instructions, reducing the training difficulty from two perspectives: first, by providing more decision-making information, and second, by shortening the prediction horizon for action sequences. Thanks to the explicit subgoal generation from the high-level visual planner, the low-level policy model is not required to master complex multi-step manipulation skills over long horizons. Furthermore, the stepwise instructions from language planning and the subgoal image chains from vision planning enable the policy model to be trained without directly conditioning the general instruction to predict the entire long-horizon action sequence. Instead, the policy model can predict action sequences in segments, effectively reducing the complexity of the policy learning task by leveraging the structured guidance provided by the language and vision planning components.

# F   More Results and Analysis of Additional Experiments

## F.1   The Effect of Encoding-side Alignment

To effectively improve the perceiving capabilities of the MLLM and lay a solid foundation for grounded reasoning and imagination, we introduce encoding-side alignment. We design a series of visual question answering (VQA) tasks to categorize the perception capabilities into five fundamental sub-skills: object recognition, color recognition, size identification, number counting, and spatial relationship understanding. For each fundamental perception capability, we design several targeted questions. The question templates are detailed in Table 6, and can be divided into two types based on the answer format: Yes/No questions and open-ended questions. We feed the questions and corresponding visual images into the MLLM for evaluation, prompting the MLLM to generate language-only answers. In this study, we use GPT-4 and compare its generated responses with the ground-truth answers and give a evaluation of the semantic similarity and correctness ranging from 1-5, with 1 being the lowest score, indicating an incorrect answer, and 5 representing the highest similarity, indicating the most accurate and contextually appropriate answer. Notably, we find that GPT-4's similarity assessments are highly accurate and closely match expert human evaluations, allowing us to directly employ it as a scoring mechanism. This approach enables automated large-scale evaluation without the need for extensive crowdsourced annotation resources, making the assessment process more efficient and cost-effective. We attempted to use GPT-4 for more fine-grained scoring (for example, scoring from 0-100), but found that the consistency with human evaluations was not as good as the 1-5 scale. For each term of fundamental perception capability, we randomly select 100 cases from the 28 tasks dataset used in the paper for evaluation. We calculate the total score and normalize it by the maximum possible score (28 task * 100 cases * 5 score) to obtain a percentage score, which is presented in Figure 16.

Table 6: Overview of five types of fundamental perception capabilities. The question templates are illustrated and can be categorized into two types: open-ended and Yes/No questions.

| Foundation Capability | Questions | Answer Type |
|---|---|---|
| Object Recognition | Can you identify all the objects on the table? | Open-ended |
| | Are there any objects that are movable on the table? | Yes/No |
| | How many different types of objects are on the table? | Open-ended |
| | Does the letter appear on the table? | Yes/No |
| Color Recognition | What are the colors of the blocks? | Open-ended |
| | Is the robot manipulating any of the red blocks? | Yes/No |
| | How many colors are there in blocks on the table? | Open-ended |
| | Are the colors of the blocks on the desktop duplicated? | Yes/No |
| Size Identification | Can you tell the size of all the blocks on the table? | Open-ended |
| | Are the blocks the same size? | Yes/No |
| | How many different-sized blocks? | Open-ended |
| | Are the blocks relocated by the robot identical in size? | Yes/No |
| Number Counting | How many blocks are on the table? | Open-ended |
| | Which item has the highest number of different objects? | Open-ended |
| | Can you identify the number of blocks in bowls? | Open-ended |
| | How many objects were moved in total in a demonstration? | Open-ended |
| Spatial Relationship | Which corners of the table have no objects? | Open-ended |
| | Are there objects stacked on top of each other? | Yes/No |
| | Which area of the table has the most objects? | Open-ended |
| | How many layers is the highest stack of objects? | Open-ended |

Our results in Figure 16 demonstrate that carefully designed captioning tasks can significantly enhance the MLLM's performance across various foundational perceiving capabilities. When we remove the perception pretraining with encoding-side alignment, we observe more frequent hallucinations and errors in the designed VQA evaluations. It is worth noting that even the ablated version of perception pretraining still shows some advantage over EmbodiedGPT, which can be attributed to the additional image generation loss. The supervision from visual aspects encourages attention to visual details and spatial information for more grounded reasoning.

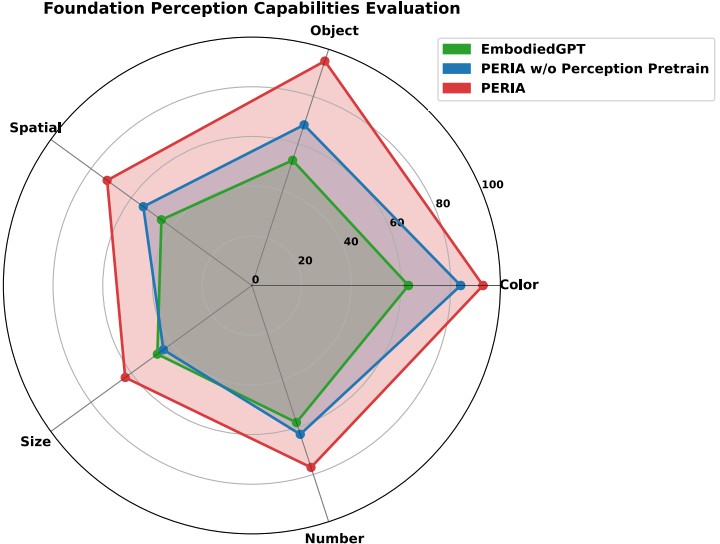

Figure 16: The evaluation of fundamental perception capabilities between language planning methods with MLLMs.

## F.2 The Effect of Joint Training

To verify the effectiveness of language planning and vision planning, we conduct an ablation study by decoupling the training of reasoning and imaging. First, we train the MLLM solely for reasoning, generating text tokens to predict stepwise instructions. Subsequently, we use only the text tokens from the stepwise instructions as conditional information to train the image editing model. It is important to note that this differs from the visual fidelity experiment mentioned in the main text, as the  tokens are entirely set to zero, and the absence of joint training eliminates the consistency loss. We introduce a semantic similarity metric to evaluate instruction following accuracy. Specifically, we calculate the CLIP similarity between generated subgoal images and general prompts, normalized by the CLIP score between the ground truth ultimate goal image and prompts. This metric reflects the progress of generated subgoal images throughout the entire chain, tracking the instruction following and gradual advancement towards ultimate goals specified by complex instructions.

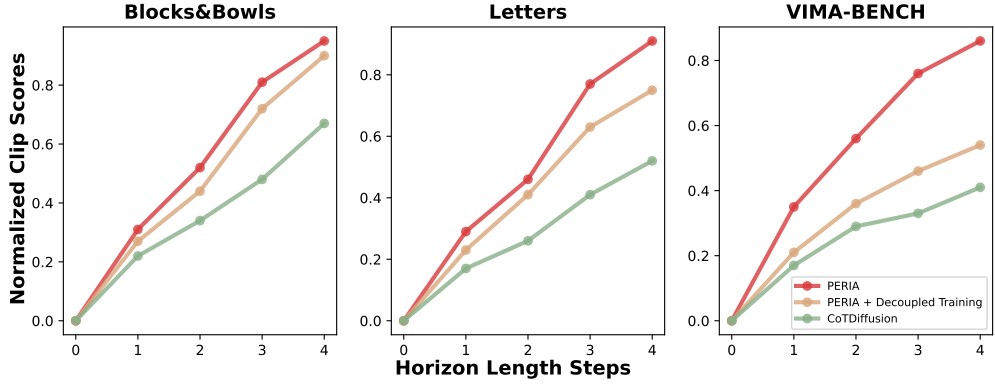

Figure 17: The evaluation of the normalized CLIP scores between instructions and generated subgoal images for each generation step, reflecting the stepwise accuracy of instruction following and the incremental progress towards ultimate goals specified by complex instructions.

To ensure a consistent comparison, we select tasks with a horizon length of 4 across all task types. The results in Figure 17 show that CoTDiffusion has the worst semantic alignment due to the lack of explicit incorporation of LLMs for logically reliable reasoning. The results reveal that the generated images from the decoupled training version exhibit relatively poor instruction following compared to the jointly trained version. We attribute this to two main reasons. First, using only text instructions as conditioned information fails to provide sufficient guidance, which can be considered as a version

with 0  tokens. Second, the absence of a consistency loss constrains and encourages semantic alignment between the generated subgoal images and instructions. In summary, the performance drop caused by decoupled training highlights the benefits of joint training, which enables a synergistic effect, more fine-grained and consistent image generation, and instruction reasoning. It is worth noting that the performance drop caused by decoupled training is more significant on the VIMA-BENCH, highlighting the importance of latent image token embeddings in providing guidance beyond language, especially in task environments where text-only instructions are challenging to describe sufficiently and completely.

## F.3 The Flexibility of LLM Backbones

To comprehensively compare the impact of different LLMs as backbones on the capabilities of the PERIA framework, we experiment with various LLMs backbones for fine-tuning, including Vicuna-7B [4], Vicuna-13B [5], LLaMA-2-7B [6], and LLaMA-3-8B [7]. The evaluation results for each model are presented in Figure 18. The results show that Vicuna-13B outperforms Vicuna-7B, indicating that larger model sizes can bring performance gains. However, the more recent and powerful LLaMA-3-8B surpasses both Vicuna models, demonstrating that our framework can achieve substantial improvements and enhancements by leveraging stronger LLM backbones.

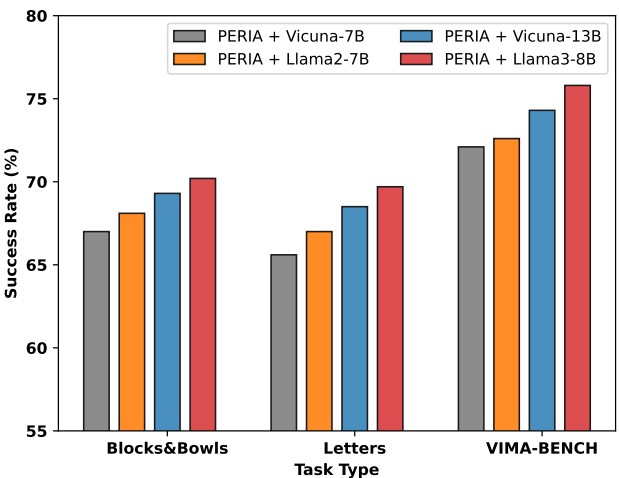

Figure 18: The evaluation of PERIA with different LLM backbones across three task types.

## G   Quick Guideline of Usage

```
from PERIA import load_peria
llm_backbone = ['Vicuna-7B', 'LLaMA2-7B', 'Vicuna-13B', 'LLaMA3-8B']
peria = load_peria(llm_backbone)
low_level_fdm = load_fdm('fdm_path')
observation = load_obs('obs_path')
task_instruction = load_ins('ins_path')
prompt = load_prompt('prompt_path')
while not done:
    stepwise_instruction = peria.language_planning(observation,
    task_instruction, prompt)
    subgoal_image = peria.vision_planning(observation, task_instruction,
    prompt)
    action = low_level_fdm(observation, stepwise_instruction,
    subgoal_image)
    observation, done = env.step(action)
```

[4] https://huggingface.co/lmsys/vicuna-7b-v1.5, 7B, version 1.5
[5] https://huggingface.co/lmsys/vicuna-13b-v1.5, 13B, version 1.5
[6] https://huggingface.co/meta-llama/Llama-2-7b-chat-hf, 7B
[7] https://huggingface.co/meta-llama/Meta-Llama-3-8B-Instruct, 8B

# H   Pesudocodes of Framework

---

**Algorithm 1** The training of PERIA for robotics manipulation.

---

**Input**: Training dataset with pair-wised keyframesstepwise sub-instructions and action trajectories, MLLM, conditional diffusion model and low-level foundation model for action planning.

---

**Perceive: Encoding-side Alignment between Vision and Language**

**for** each iteration **do**

Sample images $\mathcal{I} = \{v_1, v_2, ..., v_n\}\}$ and pairwised caption $\mathcal{C} = \{x_1, x_2, ..., x_l\}$ from the $\mathcal{D}^{train}$ as mini-batch $\mathcal{B}$

Calculate the projected visual tokens $\mathcal{W}(f = \mathcal{V}(\mathcal{I})))$ by projection layer $\mathcal{W}$ after the visual encoder $\mathcal{V}$ the encoding side.

Fed the text token from prompt and the projected visual tokens into LLM jointly and infer caption $\hat{x}_t$ in an autoregressive way:

$\hat{x}_t = \text{MLLM}(\{x_1, ...x_{t-1}\}, \texttt{prompt} \mid \mathcal{W}(f = \{\mathcal{V}(v_i)\}_{i=1}^n))$

Update parameters of $\mathcal{W}$ on mini-batch $\mathcal{B}$ according the following loss:

$\mathcal{L}_{\text{Perceive}} = \sum_{t=1}^l \text{CELoss}(\hat{x}_t, x_t)$

**end for**

---

**Reasoning and Imagine: Decoding-side Joint training for MLLM and Diffusion Model**

**for** each iteration **do**

Sample general task instructions $\mathcal{T}$, initial observation image $o$, prompt $\mathcal{P}$, pairwised sub-instructions $\mathcal{E} = \{e_1, e_2, ..., e_l\}$ and groundtruth subgoal images $\mathcal{I} = \{v_1, v_2, ..., v_n\}\}$ from the $\mathcal{D}^{train}$ as mini-batch $\mathcal{B}$

Fed the text token from general task instructions $\mathcal{T}$ and prompt $\mathcal{P}$ and the projected visual tokens from $o$ into LLM jointly and reason the stepwise sub-instruction $e'_t$: $e'_t = \text{MLLM}(\{e_1, ..., e_{t-1}\} \mid [\mathcal{P}, \mathcal{T}, \mathcal{W}(f = \mathcal{V}(o))])$, calculate the reasoning loss as follows:

$\mathcal{L}_{\text{Reason}} = \sum_{t=1}^l \text{CELoss}(e'_t, e_t)$

Extract the word embedding $w_{\texttt{[IMG]}}$ from append $\texttt{[IMG]}$ token after the reasoning phase, and extract the hidden state $h_{\texttt{[IMG]}}$ from the last layer within MLLM .

Transform $w_{\texttt{[IMG]}}$ and $h_{\texttt{[IMG]}}$ into actual visual guidance $\mathcal{U}$ via image projector $\mathcal{R}$

Generate imagined subgoal images $v$ via image editing diffusion model with conditional guidance $\mathcal{U}$, and calculate the imagine loss as follows:

$\mathcal{L}_{\text{Imagine}} = \mathbb{E}_{o,v,\mathcal{U},\epsilon \sim \mathcal{N}(0,1),t} \left[ ||\epsilon - \epsilon_\theta(z_t, t, v, \mathcal{U})||_2^2 \right] .$

Fed the generated image $v$ and the original visual observation $o$ back to MLLM and perform the same captioning task like Perceive Stage

Infer the caption $\tilde{\mathcal{E}}$ of action recognition of consequent images between $v$ and $o$

Calculate the consistency between inferred instruction $\tilde{\mathcal{E}}$ and groundtruth instruction $\mathcal{E}$:

$\mathcal{L}_{\text{Consistency}} = \sum_{t=1}^l \text{CELoss}(\tilde{\mathcal{E}}, \mathcal{E})$

Update MLLM, diffusion model $\epsilon$ and corresponding projector $\mathcal{R}$ on the decoding side:

$\mathcal{L}_{\text{Total}} = \mathcal{L}_{\text{Reason}} + \mathcal{L}_{\text{Imagine}} + \mathcal{L}_{\text{Consistency}}$

**end for**

---

**Act: Training of Goal-conditioned Low-level Policy**

**for** each iteration **do**

Sample the oracle action trajectories $a$, current observation $o$, stepwise instruction $e$ and pair-wised subgoal images $v$ from the $\mathcal{D}^{train}$ as mini-batch $\mathcal{B}$

Predict the action $\hat{a}$ with the instruction $e$ and corresponding $v$ simultaneously

Update low-level policy $\psi$ on mini-batch $\mathcal{B}$ according the following loss:

$\mathcal{L}_{\text{Act}} = \sum_{t=1}^T ||\hat{a}_t - p_\psi(a_t|o_t, e_t, x_t)||$

**end for**

---

# I   Limitation & Future Work

While PERIA demonstrates significant improvements in long-horizon manipulation tasks with complex instructions, there are still some limitations that need to be addressed in future work.

- The current implementation of PERIA relies on a pre-collected dataset for training the MLLM and diffusion model. Although this allows for effective learning of perception, reasoning, and imagination capabilities, it may limit the framework's adaptability to novel environments or tasks that deviate significantly from the training data. Future work could explore methods for online learning or adaptation to enable PERIA to generalize to new situations more effectively.
- The joint training of the MLLM and diffusion model can be computationally intensive and time-consuming, particularly when generating high-quality images. While we have demonstrated the effectiveness of this approach, further research is needed to optimize the training process and improve its efficiency. This could involve the development of more lightweight architectures, advanced training techniques, or parallelization strategies.
- While PERIA has shown promising results in simulated environments, its performance in real-world scenarios remains to be explored. Real-world manipulation tasks may introduce additional challenges, such as noisy sensory inputs, dynamic environments, and physical constraints, which could affect the framework's performance. Future work should investigate the deployment of PERIA on physical robotic systems and assess its robustness and effectiveness in real-world settings.

Despite these limitations, PERIA introduces a novel and promising paradigm towards enabling robots to perform complex manipulation tasks with general instructions. By addressing these challenges and continuing to refine the framework, we hope that PERIA can provide some insights to the robotics manipulation research towards long-horizon tasks with more complex instructions in free-form, paving the way for more intelligent and versatile robotic systems that can effectively operate in a wide range of environments and applications.

# J   Social Impact

By enabling robots to understand and follow more natural and diverse human instructions, PERIA can facilitate seamless human-robot collaboration in industries such as manufacturing, healthcare, and household assistance. This could lead to increased productivity, improved quality of life, and the creation of new job opportunities. For instance, an educational robot equipped with the PERIA framework could help children engage in constructive play activities, such as building block games or puzzles. The robot could provide step-by-step guidance and demonstrations, adapting to the child's skill level and learning pace. This interactive and personalized approach to learning could enhance children's cognitive development, problem-solving skills, and creativity.

In conclusion, the advancements in long-horizon manipulation tasks presented in this work have the potential to advance the progress in the field of intelligent embodied robots, but responsible development and deployment practices must be adopted to ensure the safe, ethical, and beneficial integration of robots in educational or industrial settings.

