# OpenReview forum: "PERIA: Perceive, Reason, Imagine, Act via Holistic Language and Vision Planning for Manipulation"
_NeurIPS.cc/2024/Conference — NeurIPS 2024 poster_

### Official Review · Reviewer_WmQe · 2024-06-20

**Soundness:** 3
**Presentation:** 3
**Contribution:** 3
**Rating:** 6
**Confidence:** 4

**Summary:**

The paper proposes a framework that integrates large multimodal language models (MLLMs) and diffusion models to enable holistic language planning and vision planning for long-horizon robotic manipulation tasks with complex instructions. The authors jointly train the MLLM and diffusion model for language reasoning and visual imagination through latent image token generation. An explicit consistency loss aligns the reasoned instructions with the imagined subgoal images.

**Strengths:**

1. Novel motivation for integrating of multiple modalities for providing better guidance.

2. Principled design of the framework components like the encoding-side alignment and the latent image token generation approach.

**Weaknesses:**

1. Weak experimental evaluation (see below questions).

**Questions:**

1. While the authors acknowledge that training and inference costs are significant, the current draft lacks a more in-depth analysis of these problems. In particular, what are the various tradeoffs associated with different MLLMs that can be used, taking into consideration training time/FLOPs/MACs? How does varying these choices impact performance? Experiments answering these questions are equally as important as the ablations being run on training design choices (e.g. alignment loss).

2. Lack of real-world evaluation. Many works ([1], [2]) in this problem setting leveraging foundation models for robotic manipulation demonstrate the advantages of these large MLLMs/generative models in real-world settings, where the distribution of objects is extremely long-tailed. Can the authors show that PERIA can operate with similar success in this regime?

[1] [Look Before You Leap: Unveiling the Power of GPT-4V in Robotic Vision-Language Planning](https://arxiv.org/abs/2311.17842)

[2] [Zero-Shot Robotic Manipulation with Pretrained Image-Editing Diffusion Models](https://arxiv.org/abs/2310.10639)

**Limitations:**

Yes, the authors address the limitations of their work.

---

> ### Author Rebuttal · Authors · 2024-08-07
>
> # Q1 More analysis of Training Cost & Performance between LLM backbone
>
> Thanks for the insightful suggestions! Considering most PERIA's computational cost is to align between vision and language, and bridge the gap between pretrained LLMs and image editing models in general domains versus robotics manipulation domain, we conduct further investigation by substituting enhanced model backbones for both the visual encoder and LLM. We compared the training time and performance until convergence and utilized CalFlops, an open-source tool designed for calculating FLOPs and MACs for neural networks, with results summarized as follows:
>
> |Model Variant|Training Time (Perception) ↓|Training Time (Reason+Imagine) ↓|FLOPs ↓|MACs ↓|Success Rate ↑|
> |--|--|--|--|--|--|
> |PERIA(ViT+Vicuna7B，reported in paper)|8 hrs|42 hrs|1815.38|907.3|68.2|
> |PERIA(ViT+Vicuna 13B)|13 hrs|66 hrs|3679.82|1810.7|71.9|
> |PERIA(ViT+llama2 7B)|8 hrs|41 hrs|**1740.89**|**874.8**|69.0|
> |PERIA(ViT+llama3 8B)|7 hrs|39 hrs|1926.17|962.2|**73.1**|
> |PERIA(LIV+Vicuna 7B)|5 hrs|38 hrs|1804.60|899.1|69.2|
> |PERIA(LLaVA1.5 7B)|**4.5 hrs**|**32 hrs**|1814.59|903.0|70.3|
>
> **Key Findings:**
>
> - **LLM Capabilities Impact Performance and Efficiency:**
>
> Stronger LLMs generally improve PERIA's performance. Replacing Vicuna-7B with models like Vicuna-13B, LLaMA2-7B, and LLaMA3-8B led to varying improvements. Training time is influenced by parameter numbers and inherent LLM capability. Despite its larger size, Vicuna-13B underperformed LLaMA3-8B in both performance and training efficiency due to higher parameters and lower capability. Conversely, LLaMA3-8B, with more parameters than Vicuna-7B, showed reduced training time, likely due to its stronger general-domain knowledge facilitating easier fine-tuning.
>
> - **Pretrained MLLM and Visual Encoder Enhances Performance and Efficiency:**
>
> Pre-trained MLLMs, such as LLaVA 1.5, which have undergone general domain vision-language alignment, obviate the need for alignment LLM and visual encoder from scratch. PERIA trained on LLaVA 1.5 can significantly reduce training time and improve the overall performance. Similarily, LIV, a robotics-specific representation with fewer parameters than ViT-B-32, leveraged its pre-training on robotics datasets to achieve vision-language alignment, alleviating further alignment efforts.
>
> **Conclusion:**
>
> Our findings suggest balancing stronger LLMs' performance benefits against the computational costs of larger parameters. We recommend prioritizing capable LLMs within similar size constraints and favoring MLLMs or models pre-aligned with the robotics domain. Future work will explore **more powerful model backbones** and **efficient fine-tuning techniques** to reduce computational costs. Additionally, we aim to investigate **lightweight subgoal modalities** (such as object masks, bounding boxes, or keypoint tracking) to balance cost and guidance. We also hope PERIA can serve as a **foundational model** for MLLM-based embodied manipulation research, offering a more efficient starting point than learning from scratch with non-robotics-tuned MLLMs.
>
> # Q2 Real Robotics Evaluation
>
> Good suggestions! We highly agree on the importance of demonstrating PERIA's capabilities in real-world settings, but we face limitations due to the absence of robotics in our lab. To address this, we use the BridgeData v2 dataset, which features real-world manipulation videos and corresponding action annotations, to evaluate PERIA's potential through:
>
> 1. **Language Planning Accuracy**: Assessing action prediction accuracy against provided annotations.
> 2. **Visual Planning Fidelity**: Evaluating instruction-following ability by generating goal images from given action annotations.
>
> This approach allows us to isolate and assess PERIA's high-level cognitive and planning abilities in real-world scenarios, excluding low-level policy execution. The results are as follows (We want to clarify that SuSIE is already included as a key baseline of visual planning method in our main paper. ViLA is also mentioned in related work section but not as baseline before rebuttal due to its unavailability of code and its similar training-free paradigm with the PAR approach baseline, differing primarily in the LLM backbone used. For this, we implement the base version of ViLA with GPT4V based on PAR for the limited timewindow):
>
> ||**Language Accuracy↑**|**Visual Fidelity↓**|
> |--|--|--|
> |ViLA|0.69|-|
> |SuSIE|-|21.2|
> |**PEIRA**|**0.76**|**17.5**|
>
> Due to the free-form nature of language annotations in BridgeData v2, we calculate language accuracy with semantic similarity rather than token accuracy, as both metrics show consistent trends in the simulator results presented in Tab.2 of main paper. The visualization example can be found at Fig.6 in uploaded PDF. More details of related work can also be found at Tab.2 in uploaded PDF.
>
> **Key Findings:**
>
> - Using the same backbone Instructpix2pix as SuSIE, PERIA generated more coherent and task-relevant images, highlighting the importance of visual tokens and the integration of MLLMs in our approach.
> - Language accuracy decreased for open-vocabulary tasks in real domains compared to simulator performance for the Sim2Real gap, but PERIA still outperforms ViLA for the enhanced perception pretraining, which improves its ability to correctly interpret and describe complex manipulation tasks in real-world scenarios.
>
> **Conclusion:**
>
> We will conduct comprehensive real-world evaluations as soon as resources permit and we may try SimpleEnv as a temporal substitute of real-robot before ready. Additionally, we will focus on collecting and incorporating more real-robot datasets for pretraining, enhancing PERIA's adaptability to real-world scenarios.
>
> ---
>
> **Sincere thanks for your insightful review!  We hope our response can address the concerns.**

---

> > ### Comment · Reviewer_WmQe · 2024-08-13
> >
> > I have read the authors' rebuttal and comments. Thank you for your detailed response. I maintain my positive rating.

---

> > > ### Author Response · Authors · 2024-08-13
> > >
> > > We sincerely thank you for your recognition of our work! We will incorporate the corresponding details into the updated version. The constructive suggestions really help us improve the quality of the paper! Please let us know if you need any further information or clarification.

---

### Official Review · Reviewer_tWdi · 2024-07-13

**Soundness:** 3
**Presentation:** 3
**Contribution:** 2
**Rating:** 6
**Confidence:** 4

**Summary:**

The paper tackles the problem of long-horizon task planning on pick-and-place tasks in the Ravens domain. Given a dataset of trajectories, it first learns the projection to align the vision and language encoder for a multimodal LLM. Then it finetunes both the multimodal LLM and a diffusion model to generate a step action in language, where the diffusion model is used to generate a conditioning subgoal image, which is proposed as an intermediate step that helps with the step action generation in language.

**Strengths:**

- The paper is overall well-written and the figures are helpful for understanding the method.

**Weaknesses:**

- It is unclear, at least from the experiments in the paper, that the diffusion model is actually useful, especially when the output is still in language space. For example, it seems that the tasks studied in the paper can be easily tackled by a modern multimodal language model (likely even the open-sourced ones), by simply providing the the initial image and appropriate prompting. However, this is missing as an important baseline in the paper (and this does not require additional training data). Furthermore, to demonstrate the effectiveness of an image subgoal in addition to a language subgoal, the evaluation would have to be done on tasks that have subgoals that are difficult to describe in language but easy to describe in visual space, but all the evaluated tasks are the contrary.
- A related work “Video Language Planning” also seems to be missing from the paper, despite it might involve closed-sourced models. However, the idea seems quite relevant and it’s unclear if the paper provides additional insights for the community.

**Questions:**

See "weaknesses" section above.

**Limitations:**

The limitations are described in the paper.

---

> ### Author Rebuttal · Authors · 2024-08-07
>
> # Q1 Effectiveness of Diffusion model &  MLLMs & Tasks better described in image
>
> Thank you for insightful questions! We appreciate the opportunity to respond point by point:
>
> 1. **Tasks better described in image**
>
>     We highly agree that tasks with subgoals better described in visual space are crucial. Our evaluation spans three task types. For Blocks & Bowls and Letters, the textures, colors, and backgrounds are relatively simple and straightforward to describe verbally. However, from VIMA-BENCH, which is specialized with multimodal instructions (interleaved image and language), we selected 8 long-horizon tasks categorized into three types:
>
>    - **Rearrange**: requires precise absolute positioning, which is challenging to specify accurately through qualitative language alone, image with subgoal location is more clear.
>    - **Constraint**: involves explicit intermediate or outcome constraints. Language instruction are often lengthy and unclear.
>    - **Follow:** infers related objects from given example and reason. Object shapes, colors, and textures are often difficult to describe verbally, especially with multiple distractors.
>
>    We believe these tasks are challenging to clearly described in language only and more suitable for image subgoals. For illustrating examples, please refer to Fig.2 in uploaded PDF.
>
> 2. **Effectiveness of Diffusion model:**
>
> - **Providing holistic planning capabilities:**
>
> Evaluations across all task types demonstrate the performance gains of holistic planning over language planning methods, with improvements of +19.4% in Blocks & Bowls and +17.4% in Letters. The advantages of holistic planning are even more pronounced in VIMA-BENCH with +37.1%, highlighting the difficulty of adequately describing subgoals using language alone, as shown in Tab.1 of main paper. PERIA's own ablation studies on subgoal modality also support similar conclusions, shown in Fig.3 in uploaded PDF. More detailed subgoal guidance always works.
>
> - **Enhancing language planning accuracy:**
>
> Diffusion model, serving as a visualization module, introduces pixel-level supervision from groundtruth subgoal images to the MLLM. This integration, along with a consistency loss, forms an intermodal bridge that enables language planning to benefit from pixel-level guidance, thus avoiding isolated learning within language supervision.Our ablation studies, which excluded diffussion model and entirely removed visual supervision from PERIA, resulted in a decrease in language planning accuracy, shown in Tab.2 of uploaded PDF. This demonstrates that image generation as an auxiliary prediction task enhances the MLLM's understanding of task instructions and scenes, thereby improving reasoning capabilities, proving the phrase "What I cannot create, I do not understand".
>
> 3. **Capability of pre-trained MLLMs:**
>
> Good idea!  We tested several popular MLLMs, including GPT4V, Claude 3.5 and InternVL-2 with web interfaces. And we conducted a comprehensive evaluation of GPT-4V key and LLaVa 1.5. The results are shown in Tab.2 of uploaded PDF.
>
> Due to domain gaps between robotics and general domain, these models often misidentified scene object properties, directly impacting reasoning, shown in Fig.5 in uploaded PDF. While GPT-4V demonstrated superior accuracy among MLLMs, it still underperformed compared to the fine-tuned EmbodiedGPT and PERIA. We do not expect to achieve superiority over pretrained MLLMs in general domain. Rather, we highlight that deploying MLLMs in robotics manipulation requires substantial domain-specific data fine-tuning and foundational capability enhancements (see LLaVa 1.5 after finetune).
>
> # Q2 Discussion with VLP
>
> Good suggestions! We now make a detailed comparison between PEIRA with VLP.
>
> **Similarities:**
>
> 1. **Motivation:** Leverage a generative model to generate subgoal with modalities beyond language to offer more sufficient guide
> 2. **Overall pipeline**:  VLM/MLLM + generative model + low-level policy
> 3. **Necessity for fine-tuning:** VLP finetunes PALM-e 12B and PERIA finetunes Vicuna 7B both using addtional robotics domain dataset.
>
> **Key Differences:**
>
> 1. **Training Paradigm:** VLP employs decoupled training for reasoning and imagination, with generated videos subject to VLM evaluation before adoption. In contrast, PERIA utilizes joint training of MLLM and diffusion model, aligning their latent spaces and employing a consistency loss to encourage semantically matching language and visual planning.
> 2. **Subgoal Modality:** VLP generate short-horizon video as subgoal, while PERIA generate keyframe images, offering a more lightweight representation.
> 3. **Low-Level Policy:** VLP offers three methods, including UniPi's inverse dynamics model and two variants of policy conditioned on either last frame or every frame of generated videos. PERIA's approach is most similar to VLP's last-frame conditioned policy, while avoiding the high fidelity and continuity requirements of the video generation, particularly the inverse dynamics approach.
>
> **Further Discussion:**
>
> The core motivation of both video and image subgoals is to provide sufficient guidance for action prediction. These subgoals can be represented through various modalities, including language, images, or videos, each offering different trade-offs between expressiveness and computational efficiency. Broadening our perspective, can we explore more lightweight modalities to describe these subgoals, such as **object&place mask, object bbox** or **keypoint tracking**?  These approaches could potentially reduce computational cost by focusing prediction on relevant areas but preserve essential semantic information for sufficient guidance to low-level policy. We think this is an intriguing direction for future work, offering opportunities to balance off between computational efficiency and guidance richness of subgoal modality.
>
> ---
>
> **Sincere thanks for your insightful review!  We hope our response can address the concerns.**

---

> > ### Author Response · Authors · 2024-08-11
> >
> > Dear Reviewer tWdi:
> >
> > Thanks again for your valuable comments and constructive suggestions, which are of great help to improve the quality of our work. We sincerely hope that our additional experiments and analysis can properly address the concerns. As the end of the discussion period is approaching, we are keen to read any further valuable feedback after reviewing our rebuttal.  If there are any further concerns, questions, or suggestions, please feel free to ask and discuss with us at any time. We are more than willing to response any of them. **Thanks to your hard work and insightful suggestions!**
> >
> > Best regards,
> >
> > Authors

---

> > ### Comment · Reviewer_tWdi · 2024-08-13
> > **Response**
> >
> > Thank you for the detailed response, and I appreciate the efforts for the additional experiments, which I think have greatly enhanced the paper. I have raised my recommendation accordingly.

---

> > > ### Author Response · Authors · 2024-08-13
> > > **Sincere Thanks for your Time and Effort!**
> > >
> > > We are deeply grateful for your recognition and invaluable feedback. The constructive suggestions have significantly helped us improve the quality of our paper, and we will incorporate the corresponding details into the updated version.
> > >
> > > Your positive recognition means a great deal to us, and we truly appreciate it. Thanks for your time and efforts again!

---

### Official Review · Reviewer_C2qN · 2024-07-14

**Soundness:** 3
**Presentation:** 3
**Contribution:** 3
**Rating:** 7
**Confidence:** 4

**Summary:**

The paper proposes a holistic vision-language planning method for long-horizon robot manipulation, by learning a multi-modal large language model (MLLM). The MLLM generates interleaved language actions and keyframe images based on language goal and the initial image. Each pair of generated language and keyframe image is used as conditioning of a learned motion policy for robot manipulation.

Based on a pretrained MLLM model, the paper first learns a projector to align visual encoding to with language on image captioning tasks tailored to robot manipulation. Then it applies instruction tuning to fine-tune the MLLM, an output projector, and a diffusion model to generate interleaved language and images. Additional, the authors propose another training objective to align the generated language and images. All large models are fine-tuned with LoRA.

On simulated robot manipulatio benchmarks, the proposed method outperforms imitation learning, language planning, and vision planning methods. The paper also systematically evaluates capabilities of the MLLM along different axes, and justifies the benefits introduced by each loss design via ablation studies.

**Strengths:**

- The paper tackles the important challenge of robot long-horizon planning. The proposed method plans jointly in the language and image space, providing rich information for the low-level policy to condition on.
- The paper exploits the capabilities of MLLM to generate language and images for robot manipulation, used with a separate low-level policy. I think this is good practice as MLLM is not naturally suitable to generate robot motion.
- The experiments are comprehensive and provide useful information on understanding the capability of the trained MLLM.
- The paper is in general well-written and easy to follow.

**Weaknesses:**

- The explanation of low-level policy is missing from the main paper. This part is very important - the MLLM outputs language and images only, and it's not clear how these modalities are bridged with robot motion.
- The contribution of the alignment loss between generated image and language is not sufficiently justified in the experiment. It will be helpful if the authors can provide the task success rate when the loss is absent.

**Questions:**

- I wonder which of the three pretraining tasks is the most important for vision-language alignment in the context of robot manipulation. It will be interesting if the authors can show some ablation studies on this.

---

> ### Author Rebuttal · Authors · 2024-08-07
>
> # Q1 Details of Low-level Policy
>
> Sorry for the confusion. Due to limited space, we placed the training details of  low-level policy in Appendix E. Thanks for bringing the attention to critical importance of this section for a comprehensive understanding of the PERIA architecture and we plan to incorporate this information into the updated version of main paper. Here we want to make more explanation about the low-level policy.
>
> - **Architecture**:  To enable multi-task learning, we adopt CLIPort, a wildly-used data-efficient end-to-end multi-task learning algorithm for Ravens that leverages an SE(2) action space. CLIPort introduces several variants and we utilize two version that conditioned on image and language respectively. Both utilize CLIP's visual and language encoders to extract embeddings, which are then fused with observation inputs via element-wise product. We directly train these variants with stepwise language sub-instructions and coherent keyframes as inputs, as **language conditioned version** and **image conditioned version** respectively. Moreover, to accommodate the simultaneous presence of stepwise language sub-instructions and coherent keyframes from vision planning and language planning, we design a variant that is conditioned on both image and sub-instruction simultaneously, noted as the **language-and-image conditioned version**. We maintain the output prediction architecture and just modifying the input processing for fair comparison. Specifically, we fuse the language sub-instruction and corresponding subgoal image through a cross-attention block comprising a 4-layer lightweight attention layer with 4 cross-attention heads. The fused embedding is combined with the observation via element-wise product, maintaining the subsequent modules of the original architecture. Detailed comparisons of the three policy architecture versions are presented in Fig. 4 of the uploaded PDF.
>
> - **Findings**:
>
>   - We investigated performance across tasks with varying horizon lengths of subgoals. Results in Fig. 5(b) in main paper demonstrate that the integration of both modalities provides more comprehensive guidance, significantly enhancing execution accuracy in long-horizon scenarios.
>   - Furthermore, we evaluated our model across diverse task types. As illustrated in Fig. 3 of the uploaded PDF, the additional guidance proves beneficial across all task categories. Notably, tasks from VIMA-BENCH with multi-modal instructions, which are challenging to be sufficient described by language only, exhibited particularly significant performance improvements with the incorporation of subgoal images. Enhancing subgoal guidance through richer modality consistently improves the semantic clarity and accuracy s across diverse task domains.
>
>   In future work, we aim to investigate more effective policy architectures as low-level policy backbones, such as diffusion policy, to further enhance the PERIA framework's performance and capabilities.
>
> # Q2 Contribution of Alignment loss
>
> Good suggestions! We conduct more analysis experiments to verify the effectiveness of alignment loss, including the lanugage accuracy, visual fidelity and success rate. The results can be found in Tab.2 in the uploaded PDF.
>
> - **The bridge between two modality**: The consistency loss, an auxiliary regularization term besides the supervision loss, serves as a bridge to enable language planning benefit from pixel-level guidance with subgoal images while allowing visual planning to be influenced by semantic-level supervision from language instructions, thus avoiding isolated learning within each modality. The ablation of the consistency loss degrades performance in both language accuracy and visual fidelity, demonstrating the effect of intermodal connection via consistency loss.
> - **Alleviate conflicts in holistic planning**: By incorporating the alignment loss, we strengthen the synergy between visual and language planning within the MLLM, mitigating the risk of the diffusion model and MLLM working in isolation.  The generalization evaluation across three levels demonstrate that the integration is particularly beneficial for unseen tasks, where each modality are more likely to output semantics in isolation, potentially resulting in semantic conflicts and task failure.
>
> # Q3 Importance of Three Pretraining Tasks
>
> Thanks for this awesome suggestions!  Our pretraining strategy comprises three tasks, each targeting specific capabilities:
>
> 1. Scene Description (SD): static understanding of single frame, including object understanding and spatial relationships.
> 2. Action Recognition (AR): dynamic understanding of subsequent images and semantic correlation between language instructions and subgoal images.
> 3. Video Understanding (VU): continuous comprehension across successive subgoal images and mitigating temporal hallucinations for long-horizon.
>
> We conducted an ablation study with variants: PERIA (w/o pretrain), PERIA (w/ SD), PERIA (w/ SD + AR), and PERIA (w/ VU) followed as:
>
> ||Language Accuracy ↑|Visual Fidelity ↓|Succees Rate(<= 8steps)↑|Success Rate(> 8steps)↑|
> |--|--|--|---|--|
> |PERIA (w/o pretrain) |80.2|16.8|55.5|42.4|
> |PERIA (w/ SD)|89.3|15.7|63.0|54.2|
> |PERIA (w/ SD + AR)|92.6|13.6|68.1|59.3|
> |PERIA (w/ VU)|84.2|15.4|61.0|57.2|
> |PERIA (w/ SD + AR + VU, Ours) |**97.6**|**12.3**|**71.2**|**66.1**|
>
> **Key Findings:**
> - SD is most important as the fundamental static comprehension task for other two pretrain tasks.
> - SD+AR achieve similar performance to the default version, indicating that the SD+AR pairing can develop certain VU capabilities. Conversely, VU as the sole pretraining task performs poorly may because it is too challenging to learn without sufficient foudnation capabilities from SD and AR.
> - Incorporating VU on top of SD+AR significantly improves the success rate for long-horizon tasks exceeding 8 subgoal steps.
>
> ---
>
> **Sincere thanks for your insightful review!  We hope our response can address the concerns.**

---

> ### Comment · Reviewer_C2qN · 2024-08-10
>
> Thank you for the rebuttal. I'm satisfied with the ablation studies that show the advantages introduced by the chosen low-level policy architecture and the alignment loss. It's also great to see how the three pretraining tasks contribute differently to the performance of PERIA. Good work!

---

> > ### Author Response · Authors · 2024-08-10
> > **Sincere thanks for the valuable recognition of our work!**
> >
> > **We sincerely thank you for your recognition of our work!**  We will incorporate the corresponding details into the updated version. The constructive suggestions really help us improve the quality of the paper! Please let us know if you need any further information or clarification.

---

### Official Review · Reviewer_HJPE · 2024-07-14

**Soundness:** 3
**Presentation:** 3
**Contribution:** 3
**Rating:** 6
**Confidence:** 5

**Summary:**

This paper focuses on robotic manipulation with complex instructions. It proposes PERIA, a framework that integrates MLLM and diffusion models to incorporate both language planning and visual planning for long-horizon language-instructed manipulation tasks. Specifically, PERIA first performs a lightweight multi-modal alignment to consolidate the multi-modal perception capabilities. Then, PERIA performs multi-modal instruction tuning, where it outputs both subgoal language descriptions and visual tokens, both of which are fed to a diffusion model to generate subgoal images. PERIA introduces an additional consistency loss between and generated subgoal image and language descriptions. Experimental results demonstrate that PERIA significantly outperforms competitive baselines.

**Strengths:**

•	This work follows a natural and reasonable pipeline to tackle the manipulation tasks with complex language instructions. Combining language planning and visual generation for manipulation is a sound approach.

•	The alignment stage empowers the overall capabilities, as demonstrated in the experimental part.

•	PERIA achieves convincing experimental results compared with previous works. The authors also conduct extensive ablative study to mine more insights.

**Weaknesses:**

•	End-to-end learning for such a large system requires considerable cost. Such a comprehensive framework may lead to powerful performances but the resources may be a limitation. This paper does not present how much resources PERIA uses or related experiments to address such potential concerns.

•	One of my concerns is that the consistency objective, which forces the MLLM to output subgoal language descriptions, may suffer from accumulative error. This is because when the generated subgoal image is not the desired image but is a natural image that can be reached within one-step action, the MLLM would learn an incorrect subgoal description.

•	More literature references and related baselines should be incorporated.

•	The ablation in visual planning lacks an experiment where PERIA generates subgoal images with either subgoal descriptions or generated visual tokens, which should reveal more insights into what leads to the improvements in visual planning.

**Questions:**

•	You generate subgoal images with subgoal descriptions and generate visual tokens. Why not use 1) subgoal descriptions and observation or 2) generated visual tokens alone? The former resembles a world model, and the latter sounds like a decoding of an imagined visual subgoal, both of which sound more natural. I guess you have tried the latter but found it was not as good as adding subgoal language.

•	What LLM do you use? It is possible that a powerful LLM accounts for superior performance to some extent. Have you compared the LLMs of different works?

**Limitations:**

Yes, the authors address the limitations at the end of the conclusion.

---

> ### Author Rebuttal · Authors · 2024-08-07
>
> # Q1 Computation resources
>
> Sorry for the ambiguity arising from distributed presentation of computational resource requirements across Appendix. The computational cost of PERIA across three primary stages: Perceive (8 V100 GPUs * 8 hours ), Reason & Imagine (8 V100 GPUs * 42 hours), and Act (single V100 GPUs * 12 hours).  For more details of model architecture and training hyperparameters, please refer to Appendix D and E in main paper.
>
> **More Analysis:**
>
> Considering most PERIA's computational cost is to align between vision and language, and bridge the gap between pretrained LLMs and image editing models in general domains versus robotics manipulation domain, we conduct further investigation by substituting enhanced model backbones for both the visual encoder and LLM. We compared the training time and performance until convergence, follows as:
> |Model|Training Time(Perception)|Training Time(Reason + Imagine)|Success Rate|
> |--|--|--|--|
> |PERIA (ViT + Vicuna 7B, reported in paper)|8hrs|42hrs|68.2|
> |PERIA (ViT + llama2 7B)|8 hrs|41 hrs|69.0|
> |PERIA (ViT + llama3 8B)|7 hrs|39 hrs|73.1|
> |PERIA (LIV + Vicuna 7B)|5 hrs|38 hrs|69.2|
> |PERIA (LLaVA1.5 7B)|4.5 hrs|32 hrs|70.3|
>
> **Key Findings:**
>
> a) **More Powerful LLM**: LLaMA-3-8B improved performance and facilitated easier training due to its superior common sense understanding.
>
> b) **Specialized Visual Encoder**: LIV, a robotics-specific representation with fewer parameters than ViT-B-32, leveraged its pre-training on robotics datasets to achieve vision-language alignment, alleviating further alignment efforts.
>
> c) **Pretrained MLLM:** Pretrained MLLMs such as LLaVA 1.5, having already undergone general domain vision-language alignment, obviate the need for fine-tuning from scratch, thus expediting adaptation to robotics tasks.
>
> **Summary：**
>
> We will continue to explore **more powerful model backbones** and **more efficient fine-tuning techniques** to alleviate the computational cost. Additionally, we are interested in investigating **more lightweight subgoal modalities** (maybe obj mask, bbox or keypoint tracking) to balance computational cost and sufficient guidance in future work. We also hope PERIA can serve as a **potential base foundation model** for other MLLM-based embodied manipulation research to reduce computational costs, offering a more efficient starting point compared to learning from scratch using MLLMs not fine-tuned with robotics data.
>
> # Q2 Consistency loss& Accumulative Error
>
> Good questions! We structure our response into three key points.
>
> - The case that generated subgoal image is a one-step goal, pointed out by the reviewer, is **Granularity Error**, which also includings repetitions, skipping, or even backtracking to visited subgoals. We expect PERIA to learn appropriate task decomposition granularity with the **supervision loss** from groundtruth labels for both language planning and visual planning. When visual and language planning produce semantically consistent errors (e.g., images and language in one step)  at the same time, the supervision loss imposes significant penalties still discouraging such outputs even when consistency loss is low.
>
> - The MLLM's subgoal description ability mainly stems from action recognition pretraining in the perceive stage, not solely from consistency loss. Instead, we utilize this ability to implement consistency loss as a self-supervised mechanism during the reason and imagine stages, encouraging semantic alignment between generated instructions and subgoal images within MLLM's unified latent space.
> - The consistency loss, an auxiliary prediction task besides the supervision loss, serves as a bridge to enable language planning benefit from pixel-level guidance while allowing visual planning to be influenced by semantic-level supervision, thus avoiding isolated learning within each modality. This intermodal connection fosters a more holistic and coherent learning process of PERIA.
>
> For more details of visual supervise guidance and consistency, please refer to Fig.5(a) in main paper and Tab.1 in the uploaded PDF.
>
> # Q3 More Related Work
>
> Thanks for the suggestions!  We additionally included comparisons based on existing pretrained MLLM methods and video planning approaches in Tab.2 of the uploaded PDF. We will add theses in the revised version of main paper.
>
> # Q4 & Q5 Condition Input for Visual Generation
>
> Awesome question!  We conduct the detailed ablation studied on three generation mechanisms with language subgoal descriptions, generated visual tokens, or a combination of both as condition. The results can be found at Fig. 1 of uploaded PDF.
>
> **Key Findings:**
>
> - More tokens as condition input for visual generation is always better but improvement gets marginal when token reachs a threshold.
> - Visual tokens only lack sufficient semantic, potentially leading to overfitting or training collapse, especially with a limited number of tokens.  Increasing the number of visual tokens can alleviate this but it is essentially equivalent to incorporating language tokens. **Discarding the inherent high-level semantics of language tokens and add more visual tokens to learn these semantics from scratch is inefficient, requiring more training time and wastefully ignoring exsited language planning output.**
> - Language tokens contains rich semantics but can also get benefit from more visual tokens with visual details that are challenging to describe accurately in language.
>
> The combinatorial fusion version can achieve comparable performance with fewer visual tokens, striking an efficient balance between semantic richness and visual precision.
>
> # Q6 Different LLMs
>
> Sorry for the confusions. We use Vicuna-7B as our LLM backbone. The reviewer's insight is correct - the powerful LLM can bring performance increase and the comparisions can be found at Q1 and the Appendix F.3.
>
> ---
>
> **Sincere thanks for your insightful review!  We hope our response can address the concerns.**

---

> > ### Comment · Reviewer_HJPE · 2024-08-13
> >
> > Thanks for your detailed response. Though my concerns have not been resolved, I appreciate the efforts in such a detailed rebuttal. I am still positive and will keep my score.

---

> > > ### Author Response · Authors · 2024-08-14
> > > **Sincere Thanks for your Time and Effort!**
> > >
> > > We sincerely thank you for recognizing our work!  We will continue to deeply investigate the role of the condition mechanism in image generation and provide more comparative visualizations of the consistency loss ablation. Additionally, we will incorporate detailed visualizations and corresponding analysis of the granularity error into the updated version. We sincerely appreciate your constructive suggestions, which have significantly helped us improve the quality of our paper!

---

### Author Rebuttal · Authors · 2024-08-07

# **General Response**

---

 **Sincere thanks to all the Reviewers for the valuable suggestions and recognition of our work!**

We sincerely appreciate all reviewers' time and efforts in reviewing our paper. We are glad to find that reviewers generally recognized our key contributions and clear presentation of our paper:

**Method:**  **Novel motivation** for integrating of multiple modalities for providing better guidance [Reviewer WmQe]. The proposed approach follows **a natural and reasonable** pipeline to tackle manipulation tasks with complex language instructions and combining language planning and visual generation is **sound** approach [Reviewer HJPE]. The method plans jointly in the language and image space and is **a good practice** [Reviewer C2qN].

**Experiments:** The paper achieves **convincing** experimental results compared with previous works and conducts extensive ablative studies to **mine more insights** [Reviewer HJPE]. The experiments are **comprehensive** and provide useful information on understanding the capability of the trained MLLM [Reviewer C2qN].

**Clearness:** The paper is generally **well-written** and **easy to follow** [Reviewer C2qN, Reviewer tWdi]. The **figures are helpful** for understanding the method [Reviewer tWdi].

We also thank all reviewers for their insightful and constructive suggestions, which helped a lot in further improving our paper. In addition to the pointwise responses below, we summarize supporting experiments added in the rebuttal according to reviewers' suggestions.

**New Experiments:**

1. Additional ablation studies on image conditional generation mechanisms, LLM backbone, consistency loss, pretraining tasks.
2. Additional comparisons with pretrained MLLM in robotics manipulation scenario.
3. Additional evaluations in real-robot datasets BridgeData v2.

---

We hope these new additions help address reviewers' concerns around computational resources, effectiveness of different components, potential usage in real-world scenarios, and comparison with existing methods. **We thank the reviewers for their time and feedback in improving the quality of our work, and we hope the revisions further highlight the contributions made. Please let us know if any clarification or additional experiments would further strengthen the paper. We would be happy to incorporate all these suggestions in the revised version.**

---

### Decision · Program_Chairs · 2024-09-25

**Decision:**

Accept (poster)

**Comment:**

The paper combines multi-modal LLMs and diffusion models to perform reasoning for table-top manipulation tasks based on the language instruction and the imagined sub-goals. The paper received unanimous acceptance ratings from the reviewers. The AC also believes the paper has valuable contributions to the community. However, it would be better to go beyond table-top manipulation in the future iterations of the work. Acceptance is recommended.